# Hygroscopicity of organic compounds as a function of organic functionality, water solubility, molecular weight and oxidation level

Shuang Han[1,2], Juan Hong[1,2], Qingwei Luo[1,2], Hanbing Xu[3], Haobo Tan[4,5], Qiaoqiao Wang[1,2], Jiangchuan Tao[1,2], Yaqing Zhou[1,2], Long Peng[1,2], Yao He[1,2], Jingnan Shi[1,2], Nan Ma[1,2], Yafang Cheng[6,7] and Hang Su[6,7]

[1] Institute for Environmental and Climate Research, Jinan University, Guangzhou, Guangdong 511443, China
[2] Guangdong-Hongkong-Macau Joint Laboratory of Collaborative Innovation for Environmental Quality, Guangzhou, China
[3] Experimental Teaching Center, Sun Yat-Sen University, Guangzhou 510275, China
[4] Institute of Tropical and Marine Meteorology/Guangdong Provincial Key Laboratory of Regional Numerical Weather Prediction, CMA, Guangzhou 510640, China
[5] Foshan Meteorological Service of Guangdong, Foshan 528010, China
[6] Minerva Research Group, Max Planck Institute for Chemistry, Mainz
[7] Multiphase Chemistry Department, Max Planck Institute for Chemistry, Mainz 55128, Germany

*Correspondence: Juan Hong (juanhong0108@jnu.edu.cn) and Nan Ma (nan.ma@jnu.edu.cn)*

**Abstract.** Aerosol hygroscopicity strongly influences the number size distribution, phase state, optical properties as well as multiphase chemistry of aerosol particles. Due to the big number of organic species in atmospheric aerosols, the determination of the hygroscopicity of ambient aerosols remains challenging. In this study, we measured the hygroscopic properties of 23 organics including carboxylic acids, amino acids, sugars and alcohols using a Hygroscopicity Tandem Differential Mobility Analyzer (HTDMA). Earlier studies have characterized the hygroscopicity either for a limited number of organic compounds using similar techniques or for particles at sizes beyond the micro-scale range or even bulk samples by other methodologies. Here, we validate these studies and extend the data by measuring the hygroscopicity of a broader suite of organics for particles with size under the submicron range that are more atmospheric relevant. Moreover, we systematically evaluate the roles of related physico-chemical properties that play in organic hygroscopicity. We show that hygroscopicity of organics varies widely with functional groups and organics with the same carbon number but more functional groups show higher hygroscopicity. However, some isomers, which are very similar in molecular structures, show quite different hygroscopicity, demonstrating that other physico-chemical properties, such as water solubility may contribute to their hygroscopicity as well. If the organics are fully dissolved in water (solubility $> 7\times10^{-1}$ g/ml), we found that their hygroscopicity is mainly controlled by their molecular weight. For the organics that are not fully dissolved in water (slightly soluble: $5\times10^{-4}$ g/ml $<$ solubility $< 7\times10^{-1}$ g/ml), we observed that some of them show no obvious water uptake, which probably due to that they may not deliquesce under our studied conditions up to 90 % RH. The other type of slightly soluble organics is moderately hygroscopic and the larger their solubility the higher their hygroscopicity. Moreover, the hygroscopicity of organics generally increased with O:C ratios, although this relationship is not linear.

# 1 Introduction

Atmospheric aerosol particles consist of numerous organic species with both anthropogenic and biogenic origins (Zhang et al., 2007; Jimenez et al., 2009; Zhang et al., 2015; Wang et al., 2018). These organic species often contribute a significant fraction to the mass of sub-micrometer aerosols, and have vital effects on air-quality and climate (McFiggans et al., 2006; Randall et al., 2007; Zheng et al., 2015). To obtain a systematic understanding of their effects, it is necessary to acquire correct information on the chemical composition and physico-chemical properties of these organics (Seinfeld and Pandis, 2016). Hygroscopicity is one of the most important physico-chemical properties and it describes the ability of particles to take up water and grow in size under sub- and supersaturated conditions (Petters and Kreidenweis, 2007). Thus, it strongly influences the number size distribution, phase state, optical properties as well as multiphase chemistry of aerosol particles (Cheng et al., 2008; Su et al., 2010; Hong et al., 2018; Tang et al., 2019).

Given the large number of organic species in atmospheric aerosols, the determination of their hygroscopicity is quite experimentally difficult. Current models normally use aggregate quantities, such as the atomic oxygen-to-carbon (O:C) ratio or the average oxidation state of organics to simply parameterize the hygroscopicity of organic species in ambient aerosols. However, recent studies show that the hygroscopicity of organic aerosols cannot be fully explained by their oxidation level and the empirical relationship between hygroscopicity and O:C might not be linear (Lambe et al., 2011; Kuwata et al., 2013; Rickards et al., 2013; Marsh et al., 2017). This suggests that this simplified approach to quantify organic hygroscopicity might be problematic and a more mechanistic understanding of the complex link between hygroscopicity and other physico-chemical properties such as molecular functionality, molecular weight and water solubility of organics should be examined.

Due to these challenges, prediction of the hygroscopicity of organic compounds sometimes relies on thermodynamic models which explicitly includes these properties, for instance molecular functionality, molecular weight, into simulations. These thermodynamic models, including the Aerosol Inorganic-Organic Mixtures Functional groups Activity Coefficients (AIOMFAC) (Zuend et al., 2008; Zuend et al., 2011), the Extended Aerosol Inorganic Model (E-AIM), and the University of Manchester System Properties (UManSysProp) (Clegg et al., 1998; Topping et al., 2016) use group contribution methods to calculate water activity for organic species of atmospheric relevance. However, involving these thermodynamic simulations in transport or climate models to predict the hygroscopicity for such a large number of organic compounds in ambient aerosols is computationally expensive. Moreover, these models, based on parameterizations from measurements, are semi-empirical, and thus need more experimental data to constrain their predictions (Suda and Petters, 2013). Particularly, when it comes to very dry conditions, these models may perform even worse and cannot capture the non-ideality of the solutions accurately (Ohm et al., 2015). Therefore, quantifying the hygroscopicity of atmospheric relevant organic species through laboratory measurements by systematically varying the type of studied organics is an intrinsic necessity. Chan et al. (2008) studied the hygroscopic properties and CCN activities of a series of dicarboxylic acids and saccharides using an electrodynamic balance (EDB) and found that the CCN activities of highly water soluble organic compounds can be well predicted by the Köhler theory. Suda et al. (2014) examined the hygroscopicity of a few synthetic organic compounds that

are atmospheric relevant but not commercially available using a CCN counter (CCNc). They found that the compounds with hydroxyl or carboxyl groups are the most hygroscopic, while the ones with nitrate or methylene are the least. Jing et al. (2016) investigated the hygroscopic properties of a series of dicarboxylic acids with levoglucosan using a HTDMA, but they mainly focused on the multicomponent interactions between organic compounds. Marsh et al. (2017) collected experimental

hygroscopicity data for 23 organic compounds by a comparative kinetics EDB (CK-EDB) to compare with thermodynamic predictions and discussed that the hygroscopicity of organic compounds with increasing branching and chain length are poorly represented by models.

All these cases discussed above shows that there is already some experimental hygroscopicity data for organics with high atmospheric abundance and relevance (Peng et al., 2001; Prenni et al., 2007; Chan et al., 2008; Lambe et al., 2011; Kuwata

et al., 2013; Marsh et al., 2017; Lei et al., 2018). However, some of these measurements were conducted using different techniques, all of which have different limitations. Measurements using a CCNc could only probe the hygroscopic properties or CCN activities at supersaturated conditions, where many compounds may already fully dissolve in water droplets. The EDB or CK-EDB approaches normally analyze the droplets in the micrometer size range, far beyond the size range of atmospherically relevant aerosols. In contrast, the HTDMA system allows direct measurement of particle hygroscopicity at

subsaturated conditions and for particles at the size from tens to a few hundreds nanometers, which is a good complement closing the gaps beyond the reaches of other techniques. Furthermore, some of the aforementioned studies using similar HTDMA systems focused on quite a small number of organics discussing only one or two properties potentially influencing the hygroscopicity, leading to a limited coverage of the experimental datasets. Thus, a general picture in understanding the observed hygroscopicity among different organic species still remains unclear.

Therefore, in this work, we extend the compositional complexity and diversity of the studied organic compounds with varying functional groups, molecular structures and other relevant physico-chemical properties. We try to form a systematic matrix of experimentally-determined HTDMA data synthesizing a large suite of organics, providing unambiguous measurements of particles at atmospherically relevant size range. Combined with these experimental data, we aim to evaluate the roles of different physico-chemical properties that play in organic hygroscopicity and gain new insight on their

limitations and applicability.

## 2 Measurements

Submicron aerosol particles were generated by nebulizing the aqueous solutions (0.1 g L$^{-1}$) of each compound using a constant output atomizer (TSI, 3076). The solutions were prepared by using ultrapure water (Millipore, resistivity $\geq$ 18.2 M$\Omega$). The physico-chemical properties of the studied 23 compounds are summarized in Table 1.

After particle generation, the particles were introduced into a custom-made HTDMA system where their hygroscopic growth factor (GF(RH)) can be measured. GF(RH) is defined as Eq. (1):

$$GF(RH, D_0) = \frac{D(RH)}{D_0},$$ (1)

where D(RH) and $D_0$ are the equilibrium mobility diameter of the particles at a given RH and under dry conditions (< 10 % RH), respectively. Figure S1 shows the schematic of the HTDMA system. The detailed schematic of the HTDMA system can be found in Tan et al. (2013). Residence time for humidication of the generated aerosols is around 2.7 seconds. Calibration of the system was performed using ammonium sulfate (AS) and the results shown in Fig. S2 display that the measured hygroscopic behaviour of AS agreed well with previous studies with the deliquescence RH around 78 %.

Swietlicki et al. (2008) summarized the potential sources of error in HTDMA measurements and concluded that the reliability of the measured data is strongly associated with the stability and accuracy of DMA2 RH as well as the accurate measurement of particle diameter by DMAs. According to Mochida and Kawamura (2004), the uncertainty in the measured GF can be calculated by Eq. (2):

$$\sqrt{\left(GF \frac{\sqrt{2}\varepsilon_{Dp}}{Dp}\right)^2 + \left(\varepsilon_{RH} \frac{dGF}{dRH}\right)^2},$$ (2)

where GF is the measured growth factor with respect to any measured RH, $\varepsilon_{Dp}$ and $\varepsilon_{RH}$ are the errors in the measured Dp and RH. In our system, the accuracy of DMA2 RH was maintained to be ±1% and the uncertainty for the mobility diameter was ±1% according to PSL (Polystyrene Latex particles) calibration. Hence, for our system, $\varepsilon_{RH}$ and $\varepsilon_{Dp}$/Dp are 1% and 0.01, respectively. The calculated uncertainty according to the above-mentioned method is added in the measured GF in the following section.

According to κ-Köhler theory, we converted the measured hygroscopic growth factor to the single hygroscopicity parameter κ (Eq. 3-6) to facilitate the comparison of the hygroscopic properties among different compounds (Petters and Kreidenweis, 2007):

$$\kappa = (GF^3 - 1)\left(\frac{Ke}{RH} - 1\right),$$ (3)

$$Ke = exp\left(\frac{4\sigma_{sol}M_w}{RT\rho_w D(RH)}\right),$$ (4)

$$RH/100 \% = a_w Ke,$$ (5)

$$\kappa = \frac{(GF^3 - 1)(1 - a_w)}{a_w},$$ (6)

where $a_w$ is the water activity, $M_w$ and $\rho_w$ are the molar mass and the density of pure water at temperature $T$, respectively; $\sigma_{sol}$ is the solution droplet surface tension, which was assumed to be the surface tension of water (0.072 J m⁻²) and $R$ is the ideal gas constant.

# 3 Results and discussion

## 3.1 Hygroscopicity of individual organics

In this section, we summarized the measured hygroscopic properties of the 23 organic species, which are classified into three groups based on their functionality. Particles at the dry size of 200 nm were selected for analysis. Comparison with previous literature data for those compounds were systematically conducted in the following section. The available information with respect to the deliquescence RH (DRH), efflorescence RH (ERH), phase transition, the measured $\kappa$ at 90% RH for each compound as well as the used instrument in different works are summarized in Table 2.

### 3.1.1 Carboxylic acids

Carboxylic acids are the most abundant water-soluble components identified in atmospheric aerosols (Chebbi and Carlier, 1996; Mochida et al., 2003; Kundu et al., 2010). Hygroscopic properties of straight-chain dicarboxylic acids have been extensively investigated in previous studies (Chan et al., 2008; Kuwata et al., 2013; Rickards et al., 2013), however, HTDMA data for dicarboxylic acids with additional substitutions and tricarboxylic acids are limited. To achieve an overview

of the hygroscopicity of carboxylic acids, we measured the water uptake of several common straight-chain dicarboxylic acids in the atmosphere and further extended the hygroscopic measurements for dicarboxylic acids with substitutions and tri-carboxylic acids. Figure 1 & 2 shows the measured humidograms of straight-chain dicarboxylic acids (Fig. 1a-f), dicarboxylic acids with substitutions (Fig. 2a-c) and three tricarboxylic acids (Fig. 2d-f), respectively.

Among the studied straight-chain dicarboxylic acids, only malonic acid showed continuous hygroscopic growth with

increasing RH and the measured GF at 90 % RH was 1.47, shown in Fig. 1a. This  agrees quite well with previous studies using other HTDMA systems or other techniques, for instance, EDB, with longer residence time for humidification (Peng et al., 2001; Prenni et al., 2001; Wise et al., 2003; Salcedo et al., 2006; Jing et al., 2016). Previous studies (Swietlicki et al; 2008; Duplissy et al., 2009; Wu et al., 2011; Suda et al., 2013) suggested that the residence time for humidification may also potentially influence the observed water uptake of particles as the measured particles, especially for some organic

compounds, may not reach their equilibrium humidified sizes during a quite short time of wetting. However, current results with good consistency with other studies of much longer residence time suggests that malonic acid may already reach equilibrium during the quite short time of humidification in our system. On the other hand, Braban et al. (2003) reported that the DRH for malonic acid ranges between 69% - 91% RH. However, within this RH range we did not observe any clear evidence for phase transition. Braban et al. (2003) also reported that the ERH for malonic acid is around $6\pm3$ %RH, which

is lower than the condition that our system was able to reach. Thus, the generated malonic acid particles after drying in our study may not crystallize and existed at liquid state, which was further confirmed by the good agreement between our results and the ones obtained from the dehydration process by Peng et al. (2001).

The other straight-chain dicarboxylic acids (i.e., succinic, adipic, suberic and azelaic acids) did not show any water uptake at RH <= 90 %, as shown in Fig. 1b-e. This non-hygroscopicity was also found for succinic acid in Peng et al. (2001), for

adipic acid in Chan et al. (2008) and Dinar et al. (2008) and for azelaic acid in Chan et al. (2008) and Cook et al. (2011), either using a different HTDMA or an EDB. It has to be noted that no previous data was available for suberic acid, though Chan et al. (2008) performed its hygroscopicity measurements but failed to reported any results due to the trapping difficulties of this compound in their system. Thus, other measurements using different methods should be considered for future comparison. For all these dicarboxylic acids with no water uptake, Chan et al. (2008) explained that they have quite

low-solubility in water and once they crystallized, they would not deliquesce even under high RH conditions (e.g., RH < 90 %). Moreover, we found that the measured GFs of azelaic and suberic acids were less than 1, which could be attributed to the adsorption of a small amount of water at the particle surface, leading to the rearrangements of the microstructure and compaction of the particle (Mikhailov et al., 2004; Mikhailov et al., 2009). For pimelic acid, we observed a slightly weak hygroscopicity (Fig. 1f), which is significantly different from the result by Chan et al. (2008) that no water uptake was

observed even at RH at 90% for this compound. Based on current comparison with limited data available, we were not be able to confirm which data is more accurate, especially as Chan et al. (2008) measured the particles at micron size range using a different technique with longer residence time. Therefore, the usage of current dataset for this compound should be more careful and repeated hygroscopicity measurements using different methods should be conducted, particularly for dealing with size-dependent hygroscopicity and the influence from residence time.

The humidograms of the three dicarboxylic acids with substitutions (i.e., double bond or hydroxy group) are illustrated in Fig. 2a-c. The continuous water uptake of maleic acid (Fig. 2a) indicates that these particles may be at liquid state under dry conditions, which agrees well with the results of Suda et al. (2013) using another HTDMA at a humidification residence time of 6s. However, the results of Choi & Chan (2002), using an EDB with humidification time of 40 minutes, show that maleic acid started to take up water at RH around 75% and this deliquescence process continued until the RH reached to 85%, after

which similar growth factor values as ours were observed. Using nonisopiestic method, both Wise et al. (2003) and Marcolli et al. (2004) reported that the DRH of maleic acid would be around 89% RH, being different from Choi & Chan (2002) using the EDB. Such a large variation in the phase state as well as the phase transition process for this compound provides a note of caution for other researchers when using different data sets. However, the good agreement in GF at 90% RH among different studies suggests that further analysis in Sect. 3.2 using GF at 90% RH should be still reliable.

Continuous water uptake was also found for malic acid and tartaric acid, similar to those measured by Peng et al. (2001) using an EDB with longer residence time or Cook et al. (2011) using a different HTDMA. However, for both compounds there was a clear and considerable difference in GF at RH lower than 80% between our study and the ones by Peng et al. (2001) during hydration process. This could be due to the short residence time of our HTDMA system, limiting the water uptake of aqueous malic acid. However, previous studies (Apelblat et al., 1995; Marcolli et al., 2004; Clegg & Seinfeld,

2006) suggested that both compounds may deliquesce during wetting process, having the DRH potentially ranging between 75% to 82% RH for malic acid and 77% to 78% RH for tartaric acid. Then, their conclusion was not consistent with the results by Peng et al. (2001), where no phase transition was reported. In our study, a small leap of the GFs from 80% to 85% RH was observed for malic acid, implying that these particles were only partially deliquesced and further dissolution

occurred at elevated RH. This, on the other hand, partly supported those studies suggesting the deliquescence of malic acid during hydration. Taking both, we are then not sure which factor, either different residence time of humidification or the partial deliquescence, should be considered to be responsible for the difference in GF at intermediate RHs and thus further evidence from other measurements is essentially needed.

Gradual water uptake upon hydration was observed for citric acid (Fig. 2d), which is consistent with other studies (Peng et al., 2001; Wang et al., 2018). However, our measured GF at any certain RH was somewhat higher than the ones by Wang et al. (2018) using another HTDMA, but lower than the ones by Peng et al. (2001) using an EDB. The residence time of wetting inside both instruments were longer than our system but caused opposite effect on the measured GFs, indicating that the influence of residence time on the observed water uptake of particles is not conclusive based on current datasets. Continuous hygroscopic growth over the studied RH range was observed for the other two tricarboxylic acids (e.g., aconitic acid and butane-1,2,4-tricarboxylic acid, Fig. 2e-f), indicating no obvious phase change for these particles upon hydration. Based on their GF values at 90% RH, both compounds can be considered as more hygroscopic substances. It has to be noted that it is, to our knowledge, for the first time to report their hygroscopicity within aerosol community. These new datasets, though also need to be further validated with other measurements, might benefit the elaboration of current data pool with extended coverage of atmospherically relevant compounds.

### 3.1.2 Amino acids

Figure 3 shows the measured humidograms of the 5 amino acids and their individual comparison with previous studies. Chan et al. (2005) reported that the DRH for alanine and glycine was 96.9% and 93.2% RH, respectively. Therefore, generally no water uptake at RH <=90% was observed for both compounds in our study as well as theirs. Particularly for glycine (see Fig. 3b), a continuous shrink in wet particle size was observed from 30 % RH to 80 % RH, and above 80 % RH the GFs increased slightly. Previous studies have reported that glycine particles started to absorb water above 60 % RH (Chan et al., 2005; Marsh et al., 2017; Darr et al., 2018) prior to deliquescence due to capillary effect, which could be the potential reason for the shrinkage in their particle size in our study as a result of the microstructural rearrangement of particles upon humidification. The sizing of these structurally-rearranged particles, especially at lower RH range, will be erroneous as the volume change of the particles upon wetting may not only due to the water absorption but also the compaction of the original particles. This phenomena complicates the accurate estimation of the actual water amount absorbed by the particles due to their intrinsic hygroscopicity. In a recent study by Nakao et al. (2014), in order to avoid the influence from particle restructuring upon wetting, they sized wet particles without drying after generation and studied their droplet activation using a wet CCN. This approach they introduced might be an easier attempt, offering an unique solution for current problem from particle restructuring during the hydration processes.

Continuous water uptake was observed for particles of aspartic acid (Fig. 3c), indicating that there was no phase transition occurred during the hydration cycle. This pattern was consistent with the ones measured by Hartz et al. (2006) but with slightly lower GF values. As Hartz et al. (2006) used an EDB to measure the hygroscopicity of aspartic acid, the potential

reason for the different GF would more likely be the different residence time of humidification between different techniques, though future data should be obtained to confirm this estimation.

Chan et al. (2005) reported that the DRH values for glutamine and serine were 98.8% and 99.1% RH and thus those compounds were considered as non-hygroscopic at RH below 90%. Our observation deviated significantly from theirs, as gradual hygroscopic growth, indicating liquid state, was found for those two compounds. Previous study (Gregson et al., 2020) pointed out that particles without sufficient drying after generation, would not crystallize, for instance, the sample RH after drying was higher than their ERH. However, Chan et al. (2005) also reported that the ERH for glutamine and serine were 72.1-74.7 % and 27.6-30.8% RH, respectively, being much higher than that of our dried aerosols. Then, this effect with respect to drying process is not the plausible reason for the continuous water uptake of both compounds in our study. Ma et al. (2020) reviewed the ERH values for a series of inorganic and organic compounds and found that particles at micron size range exhibited different ERH as compared to those of submicron particles. On this basis, we speculated that the ERH values for glutamine and serine particles at nanometer range might be different, for instance, lower than the ones reported by Chan et al. (2005) and thus crystallization of these compounds did not occur in our setup. However, the conclusion from Ma et al. (2020), on the other hand, provides a hint that the DRH values for particles at micron size range could also be different than those at nanometer range. Based on this, the DRH of glutamine and serine as well as their phase state might also be size-dependent, potentially explaining the significant difference in the hygroscopic growth between our studies and the ones in Chan et al. (2005).

### 3.1.3 Sugars and Sugar alcohols

The continuous hygroscopic growth and the measured GF values for particles of fructose and mannose (Fig. 4a-b) agree well with the results of Chan et al. (2008) during both hydration and dehydration processes. The only difference is we observed a small leap in GF at RH between 70% and 75% for both compounds. However, considering the measurements uncertainties, this phenomena was not obvious that no further inference can be given. Based on the pattern of their continuous water uptake, we considered these particles exist as liquid state at dry conditions, confirmed more by that no DRH or ERH values were found for those compounds in previous studies. For sucrose (Fig. 4c), at intermediate RHs, our measured GF were lower than the ones reported by Estillore et al. (2017) using a different HTDMA with a much longer residence time (40s) as well as the ones by Hodas et al. (2015) based on DASH-SP (Differential Aerosol Sizing and Hygroscopicity Spectrometer Probe) with a residence time of 4s. And the measured GF among different studies show a strong dependence with the residence time of different instruments. As a viscous compound, the transport of water molecules within the particles as well as the particles to reach equilibrium with respect to their surrounding humid air might be quite slow. Thus, the short residence time inside the HTDMA may strongly limit the water uptake for aqueous sucrose. This conclusion was reinforced by the result of Rickards et al. (2015), who measured the water transport timescale in sucrose aerosols using aerosol optical tweezers. Water transport within such a viscous particle remained far from equilibrium even after more than 24-hours' waiting. They also suggested that the diffusion of water or dissolution of sucrose was much faster at high RH levels. We

notice, our measured GF at high RH conditions, e.g., 90%RH, was quite close to theirs, suggesting equilibrium at this point (i.e., 90%RH) might be already reached or the influence of residence time on the measured hygroscopicity is largely reduced. Therefore, analysis in Sect. 3.2 using GF at 90% RH should still be sound.

The results for xylitol and arabitol are qualitatively consistent with the ones measured by Bilde et al. (2014), though they presented their results in an AGU abstract without detailed explanation or validation. According to the measured GF, both compounds were considered as more hygroscopic and likely existed as liquid state under dry conditions. Mannitol, with GF less than unity at 90%RH from the measured humidogram, can be considered as non-hygroscopic. This was also suggested by previous literature (Ohrem et al., 2014; Martău et al., 2020). However, no practical data were presented in these works and thus only our measured data was illustrated in Fig. 4f.

### 3.2 Relating the hygroscopicity of organic compounds to their physico-chemical properties

In this section, we explore the effects from different physico-chemical properties such as molecular functionality, water solubility and organic oxidation level that potentially contribute to the observed hygroscopicity $\kappa$. According to the comparison with literature data available in Sect. 3.1, our measured growth factor at 90 % RH mostly agrees with or is close to the ones in previous works, therefore, the hygroscopicity parameter $\kappa$ discussed in this section was converted by using growth factor data measured at 90 % RH.

### 3.2.1 $\kappa$ vs. organic functionality

Figure 5a shows the measured hygroscopicity of the 23 organics as a function of carbon number. The functional groups with their corresponding numbers are indicated with colors and symbols. In order to facilitate the comparison of the compounds with the same carbons, the carbons with only one compound are not illustrated. For the studied organic compounds with the same carbon number, the hygroscopicity was increased by the addition of extra functional groups to the carbon backbone. For instance, maleic, malic, tartaric and aspartic acid with extra functional groups (e.g., C=C, -OH and -NH2) with respect to succinic acid with only two -COOHs are more hygroscopic. For C7 compounds, adding an carboxylic acid group to the carbon backbone leads to an elevated hygroscopicity from pimelic to tricarboxylic acid. Moreover, organic compounds with the same carbon numbers but different molecular functionality presented quite distinct hygroscopicity. For example, for C3 compounds, if replacing the -CH3 with an -OH or replacing the -OH group by an -COOH in their parental molecules, the hygroscopicity was significantly increased. Taking another example from C4 compounds, the organics with a hydroxyl group (-OH) instead of an -NH2 or with a double bond (C=C) instead of the hydroxyl group in their carbon backbones were more hygroscopic. Similar difference in hygroscopicity was also observed between aconitic acid (C6) with a C=C and citric acid (C6) with a (-OH). By summarizing the results in current study, $\kappa$ increased with the functionality in the following order: (-CH3 or -NH2) < (-OH) < (-COOH or C=C or C=O). However, it has to be noted that this comparison is quite qualitative, might be ambiguous and further evidence from other organic compounds is needed in order to drive a more general conclusion. Suda et al. (2012) and Chen et al. (2019) concluded that the hygroscopicity of organic compounds is closely

related to their individual polarity and highly polar compounds are usually more hygroscopic. Kier (1981) ranked the polarity of different functional groups in the sequence of -CH3< -NH2< -OH< -CHO< -NH2OH< -COOH, which could explain the difference in the hygroscopicity of organics with various functionalities in our study.

Figure 5b shows that the measured hygroscopicity of the straight-chain dicarboxylic acids alternate with the parity of the carbon numbers. It has to be noted that data of glutaric acid (C5) is quoted from Chan et al. (2008). Bilde et al. (2003) observed an alternation in the volatility of dicarboxylic acids with the number of carbon atoms similar to the ones we observed for their hygroscopicity. They attributed this to the alternation in the molar enthalpies of fusion of those compounds. Moreover, we observed that some compounds (xylitol vs. L-arabitol and fructose vs. mannose) share the same molecular formular or functionality but vary differently in hygroscopicity as shown in Fig. 5c. Both findings suggest that other physico-chemical properties of organics besides molecular functionality may also contribute to the observed variation in their hygroscopicity. Previous studies (Marcolli and Peter, 2005; Petters et al., 2017) reported that the position of the functional groups could influence the hygroscopicity properties of organic compounds. For instance, Petters et al. (2017) suggested that organic molecules with the hydroperoxyl group close the end of carbon chain were more hygroscopic. Similarly, fructose observed in our study, with the hydroxyl group in the tail of the carbon chain and being far away from the C=O group, is more hygroscopic than mannose of which these two groups are much closer to each other.

### 3.2.2 κ vs. water solubility and molar volume

Previous studies suggested that for highly soluble compounds which are fully dissolved in the aqueous droplet, their hygroscopicity are mainly controlled by their molar volume ($M_{org}/\rho_{org}$); while for slightly soluble compounds, their hygroscopicity is limited by their low water solubility (Petters et al., 2009; Kuwata et al., 2013; Nakao, 2017; Wang et al., 2019). Hence, we considered two regimes in our study: (A) compounds that fully dissolved (highly soluble with solubility > $7 \times 10^{-1}$ g/ml in this work or not saturated regime) and (B) compounds that are not fully dissolved (slightly or sparingly soluble compounds with solubility in the range between $1e^{-3}$ to $3e^{-1}$ g/ml or saturated regime) in the aqueous droplets under 90 % RH condition. In regime A, as shown in Fig. 6, the hygroscopicity decrease with increasing molar volume. Besides molar volume, the van't Hoff factor (i), which accounts for the degree of dissociation of a compound in water, could also contribute to the overall hygroscopicity for fully dissolved compounds. Sugars, as non-electrolytes with van't Hoff factor of 1, do not dissociate in aqueous solutions (Giebl et al., 2002; Koehler et al., 2006; Rosenørn et al., 2006) and thus are less hygroscopic than the dicarboxylic acids which can dissociate in water and contribute to the reduction in water activity. Frosch et al. (2010) related the van't Hoff factor with the pKa values for a series of carboxylic acids and found that the stronger the acid with smaller values of pKa, the larger the van't Hoff factor. This could explain why maleic acid, even with a larger molar volume but a smaller pKa value (1.8) is more hygroscopic than malonic acid (pKa = 2.4).

Organic compounds with low water solubility (regime B) could be obviously divided into two categories according to their hygroscopicity. One is non- or almost non-hygroscopic organics with κ close or equal to 0. These organics might present at solid or crystalline state and did not deliquesce at our measurement conditions during the whole RH range. Thus, their

hygroscopicity is not only limited by their low water solubility but also their phase state and the energy that needed for the phase transition. Compared to these non-hygroscopic slightly/sparingly soluble organic compounds, there are some other slightly/sparingly soluble organics, showing moderately water uptake with κ values larger than 0.1. These organics with limited solubility may already partially deliquesce under our studied RH conditions (Hartz et al., 2006; Chan et al., 2008), and we found that their hygroscopicity increase with water solubility. This is physically reasonable that the aqueous droplet

of these organics with limited solubility can be considered as being composed of an effectively insoluble core with a saturated solution. The organic with higher water solubility would dissolve more and have a higher molar concentration in the saturated solution. The higher molar concentration corresponds to a stronger reduction in water activity, which would lead the particles to become more hygroscopic.

### 3.2.3 κ vs. O:C ratio

Previous studies have suggested that the hygroscopicity parameter of organic species ($\kappa_{org}$) is closely related with their O:C ratios (Jimenez et al., 2009; Chang et al., 2010; Massoli et al., 2010; Cappa et al., 2011; Lambe et al., 2011; Kuwata et al., 2013; Rickards et al., 2013). In this study, we plotted our measured κ of the 23 organic compounds with their O:C ratios in Fig. 7, and for a wider atmospheric implication we compared them against previous results obtained from different atmospheric environments (Mei et al., 2013; Wu et al., 2013; Hong et al., 2015; Wu et al., 2016; Deng et al., 2018; Hong et

al., 2018; Kuang et al., 2020). Clearly, ambient organics show much lower O:C value as seen in Fig. 7. Ng et al. (2010) compiled the measured O:C data from different environments and concluded that at most sites, ambient organic aerosols mainly consist of oxygenated organic material (OOA) and hydrocarbon-like organic material (HOA). HOA, which arises from vehicle emissions, is the least oxidized with the average O:C value less than 0.2 (Ng et al., 2010; Xu et al., 2015; Xu et al., 2016; Cao et al., 2019; Li et al., 2020). Hence, with the inclusion of HOA in ambient aerosols, the average O:C value of

the bulk organic is less than 1, being generally lower than our laboratory-generated aerosols.

A general trend of the increase of $\kappa_{org}$ with increasing O:C has also been observed for laboratory results but the correlation between κ and O:C falls into two categories. One is a non-hygroscopic organic group with a weak O:C-dependence as the blue shaded area in Fig. 7. We suggested these compounds with limited water solubility might not deliquesce yet under 90 % RH as discussed previously. The other slightly/sparingly soluble organics shaded in red area in Fig. 7 is a moderate-

hygroscopic group with a slightly stronger O:C-dependence. However, the correlation of both categories is not good, which may be effected from the other properties which discussed above. Compared to those laboratory-generated pure organic compounds, ambient organics are more complex, with divergent O:C-dependent hygroscopicity among different environments. For instance, the hygroscopicity of urban aerosols in Beijing was almost constant, being less sensitive to the variations of the organic oxidation level, which is similar to our non-hygroscopic organics (Wu et al., 2016). On the contrast,

the suburban aerosols in central Germany (Wu et al., 2013) and in Guangzhou (Hong et al., 2018) exhibit a slightly stronger influence from their O:C ratio, being close to the behaviour (slope ≈ 0.12) of the moderate hygroscopic organics with relative higher water solubilities in our study. As discussed in previous works (Rickards et al., 2013) some of the laboratory-

generated pure organics share identical O:C ratio but differ widely in hygroscopicity. However, no molecular-specific information could be concluded further for those ambient organics. This, on the other hand, indicates that great uncertainties may arise from the approximation of organic hygroscopicity based on their atomic O:C ratio for ambient aerosols. The use of a simplified average property (i.e., O:C ratio) to describe the hygroscopicity of ambient organics, whose constitute may be complex, is quite risky as compounds with similar O:C ratio may vary considerably in hygroscopicity. Additional measurements of other properties (e.g., functionality or water solubility) may be difficult due to both the highly complex mixture of ambient aerosols and technique limitations. However, laboratory-generated surrogate mixtures representing the complexity of ambient aerosols at least should be examined to test the variety in the relationship between the O:C ratio and $\kappa$.

## 4 Instrument limitations

Our laboratory observations show some deviations with previous literature data measured using similar techniques with different residence time or a completely different method. The residence time of humidification inside the instrument were suggested to be one of the reason responsible for this deviation. We noticed that some compounds with a longer time of humidification show higher GF values, for instance, for sucrose at intermediate RHs. This is likely due to the kinetic limitations to water uptake, which is particularly important for viscous aerosols. This discrepancy further indicates the limitation of current method, which extra care should be taken in the future and viscosity information should always be taken into account when dealing with hygroscopicity data. However, for some other compounds, the techniques with longer residence time not always obtained the higher GF values. For instance, using an STXM (Scanning Transmission X-ray Microscopy) with a residence time much longer than our HTDMA, Piens et al. (2016) obtained a lower GF of fructose as compared to ours, which should not be caused by the evaporation losses due to its low volatility. Moreover, for glutamine and serine particles, no deliquescence was observed by Chan et al. (2005) with a residence time of as longer as 40 minutes, while in our study a moderate water uptake for both compounds were observed. These evidences pointed out, for a large suit of different compounds with various complexity, the influence of residence time on the observed water uptake of particles might not be conclusive. We may need to better understand the kinetics of water transport within particles and figure out which factors would introduce variability, for instance, measurement timescale, the choice of compounds and environmental conditions, before we design the measurement targeting the effect of residence time on aerosol hygroscopicity and eventually draw the conclusion.

The different phase transition observed between our study and the ones using the EDB (Chan et al., 2005), for instance, for glutamine and serine, reveals another limitation of current instrument in this study. One is that drying process might not be sufficient. The inadequate drying may not be able to crystallize the generated aerosols if their ERH values are quite low. This will interfere the generation of a correct phase state we are requiring. The other one is that the condition at higher relative humidity, e.g., >90%, is difficult to access. We notice that many organic compounds have their DRH higher than 90%RH and thus measurements at those conditions are quite scarce. Therefore, different means, including both raising the drying

capacity and making the elevated RH condition achievable, are all we should consider to break these limitations and improve the system.

These deviations, on the other hand, also reveal that the measurement for size-resolved particles, particularly relevant for atmospheric conditions, is still needed. As we notice, many of these previous literature data were obtained from bulk samples or particles at micron size range. Previous studies pointed out thermodynamic data of bulk samples or particles at

micron-size range may not always represent the ones at nanometre range, particularly in the size range of the atmospheric aerosol accumulation mode (e.g., ~ 102 nm). For instance, Cheng et al. (2015) suggested that particle size can be an important factor influencing the solid-liquid equilibrium upon phase transition. Pöschl et al. (2015) and Reid et al. (2018) reported that thermodynamic properties of aerosols are strongly related to particle size, for instance, viscosity or diffusion coefficient. This could be another potential reason responsible for the deviation between ours and previous

literature data. On the other hand, Rickards et al. (2015) concluded that the timescale of equilibrium for the water transport inside particles is also size-dependent. Larger particles need much more time to reach equilibrium than smaller particles. Then, the aforementioned objective, investigating the effect of residence time on aerosol hygroscopicity, should also take into account of the influence from particle size. Therefore, different approaches with capabilities accessing different particle size ranges, requiring different sample volumes, are still essentially needed.

In the previous section discussing the role of different physico-chemical properties of organics plays in aerosol hygroscopicity, the parameter $\kappa$ was converted by the measured GF under 90 % RH. For those low or sparingly soluble organic compounds, the GF−derived $\kappa$ (also known as apparent $\kappa$) is RH-dependent (see Fig. S3 as an example) and cannot express their intrinsic $\kappa$ (expressed by fully dissolved compounds), when compounds are sufficiently soluble in water. As the RH increases, further dissolution of these organic compounds with promoted hygroscopicity is expected. In the real

atmosphere, different RH conditions including both sub- and supersaturation can be reached. The measured GF or the apparent $\kappa$ of ambient aerosols at a certain RH may not be able to reveal their real hygroscopicity under various atmospheric conditions. Further calculations of other variables, for instance the liquid water content (LWC), surface area of wet particles and number concentration of CCN associated with the apparent $\kappa$ will be significantly biased. If possible, hygroscopicity measurements over large saturation range up to supersaturation, especially with the combination of CCN measurements

provide an option to reduce the uncertainties. Therefore, combination of different instruments  though with different limitations, should not be discarded but innovated.

**Data availability.**

The details data can be obtained from the corresponding author upon request.

**Supplement.**

A detailed description of the HTDMA implementation, hygroscopicity parameter $\kappa$ as a function of RH for sparingly soluble organics.

**Author contributions.**

SH did the data analysis, plotted the figures and wrote the original draft. JH collected the data, planned the study, wrote and reviewed the manuscript. NM planned the study and reviewed the manuscript. HBX and HBT contributed to the instrumentation, JCT, YQZ, LP, YH contributed to the discussion of the results. QQW, YFC and HS contributed to fund acquisition. QWL and JNS contributed to data collection and data analysis.

**Competing interests.**

The authors declare no competing financial interest.

**Acknowledgments.**

This work was supported by the National Natural Science Foundation of China (No.42175117 and 41877303), Special Fund Project for Science and Technology Innovation Strategy of Guangdong Province (No.2019B121205004), and Guangdong Innovative and Entrepreneurial Research Team Program (No.2016ZT06N263).

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

**Table 1. Substances and their relevant properties investigated in this study.**

| Compounds | Molecular structure | Chemical formula | Molar weight (g mol$^{-1}$) | Density (g cm$^{-3}$) | O/C | Solubility (g ml$^{-1}$) | Supplier, purity |
|---|---|---|---|---|---|---|---|
| Ammonium sulfate | | (NH$_4$)$_2$SO$_4$ | 132.14[a] | 1.77[a] | - | 0.77[a] | Macklin, 99.99% |
| Sugars | | | | | | | |
| D(-)-fructose | | C$_6$H$_{12}$O$_6$ | 180.16[a] | 1.59[a] | 1 | 3.75[a] | Sigma Aldrich, ≥99% |
| D(+)-mannose | | C$_6$H$_{12}$O$_6$ | 180.16[a] | 1.54[a] | 1 | 2.48[a] | Sigma Aldrich, ≥99% |
| Sucrose | | C$_{12}$H$_{22}$O$_{11}$ | 342.30[a] | 1.58[a] | 0.9 | 2.1[b] | Sigma Aldrich, 99% |
| Sugar alcohols | | | | | | | |
| Xylitol | | C$_5$H$_{12}$O$_5$ | 152.15[a] | 1.52[a] | 1 | 0.642[b] | Sigma Aldrich, ≥99% |
| L-(-)-arabitol | | C$_5$H$_{12}$O$_5$ | 152.15[a] | 1.15[a] | 1 | 0.664[d] | Sigma Aldrich, ≥98% |
| D-mannitol | | C$_6$H$_{14}$O$_6$ | 182.17[a] | 1.52[a] | 1 | 0.216[b] | Sigma Aldrich, ≥99.0% |

Carboxylic acids

| Name | Structure | Formula | | | | | Source |
|---|---|---|---|---|---|---|---|
| Malonic acid | | $C_3H_4O_4$ | 104.06[a] | 1.62[a] | 1.33 | 0.763[b] | Sigma Aldrich, 98% |
| Succinic acid | | $C_4H_6O_4$ | 118.09[a] | 1.19[a] | 1 | 0.0832[b] | Sigma Aldrich, ≥99.0% |
| Adipic acid | | $C_6H_{10}O_4$ | 146.14[a] | 1.36[a] | 0.67 | 0.03[b] | Sigma Aldrich, 99% |
| Pimelic acid | | $C_7H_{12}O_4$ | 160.17[a] | 1.33[a] | 0.57 | 0.05[b] | Yuanye Bio-Technology, 98% |
| Suberic acid | | $C_8H_{14}O_4$ | 174.19[a] | 1.30[a] | 0.5 | 0.0006[a] | Yuanye Bio-Technology, 99% |
| Azelaic acid | | $C_9H_{16}O_4$ | 188.22[a] | 1.03[a] | 0.44 | 0.0024[a] | Yuanye Bio-Technology, 98% |
| Maleic acid | | $C_4H_4O_4$ | 116.07[a] | 1.59[a] | 1 | 0.79[a] | Aladding, ≥99.0% |
| DL-malic acid | | $C_4H_6O_5$ | 134.09[a] | 1.61[a] | 1.25 | 0.592[b] | Sigma Aldrich, ≥99% |
| Tartaric acid | | $C_4H_6O_6$ | 150.09[a] | 1.79[b] | 1.5 | 1.43[e] | CATO, 99.7% |
| cis-Aconitic acid | | $C_6H_6O_6$ | 174.11[a] | 1.66[a] | 1 | 0.4[d] | Sigma Aldrich, ≥98% |
| Citric acid | | $C_6H_8O_7$ | 192.12[a] | 1.54[a] | 1.17 | 0.383[b] | Sigma Aldrich, ≥99.5% |
| Butane-1,2,4-tricarboxylic acid | | $C_7H_{10}O_6$ | 190.15[a] | 1.48[a] | 0.86 | 0.3897[c] | Bidepharm, 97% |

Amino acids

| | | | | | | | |
|---|---|---|---|---|---|---|---|
| DL-alanine | | $C_3H_7NO_2$ | 89.09[a] | 1.42[a] | 0.67 | 0.164[b] | Macklin, 99% |
| Glycine | | $C_2H_5NO_2$ | 75.07[a] | 1.59[a] | 1 | 0.25[a] | Sigma Aldrich, ≥99.0% |
| L-aspartic | | $C_4H_7NO_4$ | 133.10[a] | 1.66[a] | 1 | 0.005[a] | Sigma Aldrich, ≥99% |
| L-glutamine | | $C_5H_{10}N_2O_3$ | 146.14[a] | 1.47[a] | 0.6 | 0.0413[b] | Sigma Aldrich, ≥99.5% |
| L-serine | | $C_3H_7NO_3$ | 105.09[a] | 1.60[a] | 1 | 0.425[b] | Sigma Aldrich, ≥99% |

[a] https://www.chemicalbook.com/ [b] https://pubchem.ncbi.nlm.nih.gov/ [c] https://www.chemspider.com/

[d] https://hmdb.ca/ [e] Peng et al. (2001)



**Table 2. Summary of the measured properties with respect to their DRH, ERH, phase transition, hygroscopicity parameter (90%RH) obtained in this work and previous literature.**

| Organic groups | Compounds | DRH (%) | ERH (%) | Phase transition | κ (90% RH) | Instruments | References |
|---|---|---|---|---|---|---|---|
| Dicarboxylic acids | Malonic acid | n. o. | - | n. o. | 0.26 | HTDMA | This work |
| | | n. o. | n. o. | n. o. | 0.29 | EDB | Peng et al., (2001) |
| | | n. o. | <5 | n. o. | 0.52 | HTDMA | Prenni et al., (2001) |
| | | 69-91 | 6 | n. o. | - | AFT-FTIR; SMC | Braban et al., (2003) |
| | | 71.9 | - | n. r. | 0.25 | Nonisopiestic method | Wise et al., (2003) |
| | | 70.9 | - | n. r. | 0.28 | Nonisopiestic method | Salecedo et al., (2006) |
| | | n. o. | - | n. o. | 0.39 | HTDMA | Jing et al., (2016) |
| | Succinic acid | n. o. | n. o. | n. o. | 0.002 | HTDMA | This work |
| | | >90 | 55.2-59.3 | n. o. | 0.002 | EDB | Peng et al., (2001) |
| | Adipic acid | n. o. | - | n. o. | -0.003 | HTDMA | This work |
| | | >90 | >85 | n. o. | $2.71\ e^{-4}$ | EDB | Chan et al., (2008) |
| | | 98.5 | 98.5 | n. o. | $4.52\ e^{-4}$ | HTDMA | Dinar et al., (2008) |
| | Azelaic acid | n. o. | - | n. o. | -0.007 | HTDMA | This work |
| | | >90 | >85 | n. o. | 0.003 | EDB | Chan et al., (2008) |
| | | 81 | - | n. o. | - | HTDMA | Cook (2011) |
| | Suberic acid | n. o. | - | n. o. | -0.03 | HTDMA | This work |
| | | >90 | >85 | n. o. | - | EDB | Chan et al., (2008) |
| | Pimelic acid | take up water>85 | - | n. o. | 0.03 | HTDMA | This work |
| | | >90 | 51.5-53 | n. o. | 0.092 ± 0.009 | EDB | Chan et al., (2008) |

| | | | | | | | |
|---|---|---|---|---|---|---|---|
| Dicarboxylic acids with substitutions | Maleic acid | n. o. | - | n. o. | 0.30 | HTDMA | This work |
| | | 71-86 | 48-51 | Observed | 0.28 | EDB | Choi and Chan (2002) |
| | | 88.9 | - | n. o. | 0.27 | Nonisopiestic method | Wise et al., (2003) |
| | | 89.1 | - | n. r. | - | Nonisopiestic method | Marcolli et al., (2004) |
| | | n. o. | - | n. o. | 0.23 | HTDMA | Suda et al., (2013) |
| | DL-malic acid | a leap at 80-85 | - | Possible partial deliquescence | 0.22 | HTDMA | This work |
| | | | - | n. r. | - | Nonisopiestic method | Apelblat et al., (1995) |
| | | 75-82 | n. o. | n. o. | 0.19 | EDB | Peng et al., (2001) |
| | | n. o. | - | n. r. | - | Nonisopiestic method | Marcolli et al., (2004) |
| | | 80.5 | - | n. r. | - | Nonisopiestic method | Clegg and Seinfeld (2006) |
| | | 78.6 | | | | | |
| | Tartaric acid | n. o. | - | n. o. | 0.22 | HTDMA | This work |
| | | 77-78 | - | n. r. | - | Nonisopiestic method | Apelblat et al., (1995) |
| | | n. o. | n. o. | n. o. | 0.19 | EDB | Peng et al., (2001) |
| | | n. o. | - | n. r. | - | HTDMA | Cook (2011) |
| Tricarboxylic acids | Citric acid | n. o. | - | n. o. | 0.18 | HTDMA | This work |
| | | n. o. | n. o. | n. o. | 0.18 | EDB | Peng et al., (2001) |
| | | >90 | - | n. o. | 0.20 | HTDMA | Wang et al., (2018) |
| | Cis-aconitic acid | n. o. | - | n. o. | 0.21 | HTDMA | This work |
| | Butane-1,2,4-tricarboxylic acid | n. o. | - | n. o. | 0.14 | HTDMA | This work |

| | | | | | | | |
|---|---|---|---|---|---|---|---|
| Amino acids | DL-alanine | n. o. | - | n. o. | -0.009 | | HTDMA | This work |
| | | 96.9 | 67.3-70.8 | n. o. | $6.11\ e^{-4}$ | | EDB | Chan et al., (2005) |
| | Glycine | n. o. | - | n. o. | 0.04 | | HTDMA | This work |
| | | 93.2 | 53.6-55.2 | n. o. | 0.04 | | EDB | Chan et al., (2005) |
| | | 60 | <35 | not clear from data | - | | ATR-FTIR | Darr et al., (2018) |
| | L-Aspartic | n. o. | - | n. o. | 0.14 | | HTDMA | This work |
| | | n. o. | n. o. | n. o. | 0.18 | | EDB | Hartz et al., (2006) |
| | L-Glutamine | n. o. | - | n. o. | 0.16 | | HTDMA | This work |
| | | 98.8 | 72.1-74.7 | n. o. | 0.006 | | EDB | Chan et al., (2005) |
| | L-Serine | n. o. | - | n. o. | 0.19 | | HTDMA | This work |
| | | 99.1 | 27.6-30.8 | n. o. | 0.002 | | EDB | Chan et al., (2005) |
| Sugar and sugar alcohols | Fructose | n. o. | - | n. o. | 0.23 | | HTDMA | This work |
| | | n. o. | n. o. | n. o. | 0.18 ± 0.017 | | EDB | Chan et al., (2008) |
| | Sucrose | n. o. | - | n. o. | 0.11 | | HTDMA | This work |
| | | >90 | - | n. o. | 0.11 | | DASH-SP | Hodas et al., (2015) |
| | | n. o. | n. o. | n. o. | 0.11 | | HTDMA | Estillore et al., (2017) |
| | | - | - | n. r. | 0.10 | | HTDMA | Dawson et al., (2020) |
| | Mannose | n. o. | - | n. o. | 0.14 | | HTDMA | This work |
| | | n. o. | n. o. | n. o. | 0.183 ± 0.017 | | EDB | Chan et al., (2008) |
| | Xylitol | n. o. | - | n. o. | 0.21 | | HTDMA | This work |

| | | | | | | |
|---|---|---|---|---|---|---|
| | n. o. | - | n. o. | 0.18 | HTDMA | Bilde et al., (2014) |
| L-arabitol | n. o. | - | n. o. | 0.19 | HTDMA | This work |
| | n. o. | - | n. o | - | HTDMA | Bilde et al., (2014) |
| D-mannitol | n. o. | - | n. o. | -0.01 | HTDMA | This work |

κ values were derived from hydration data at around of 90% RH.

n. o. refers to no crystallization was observed.

n. r. refers to not reported in the work.

- refers to not measured in the work

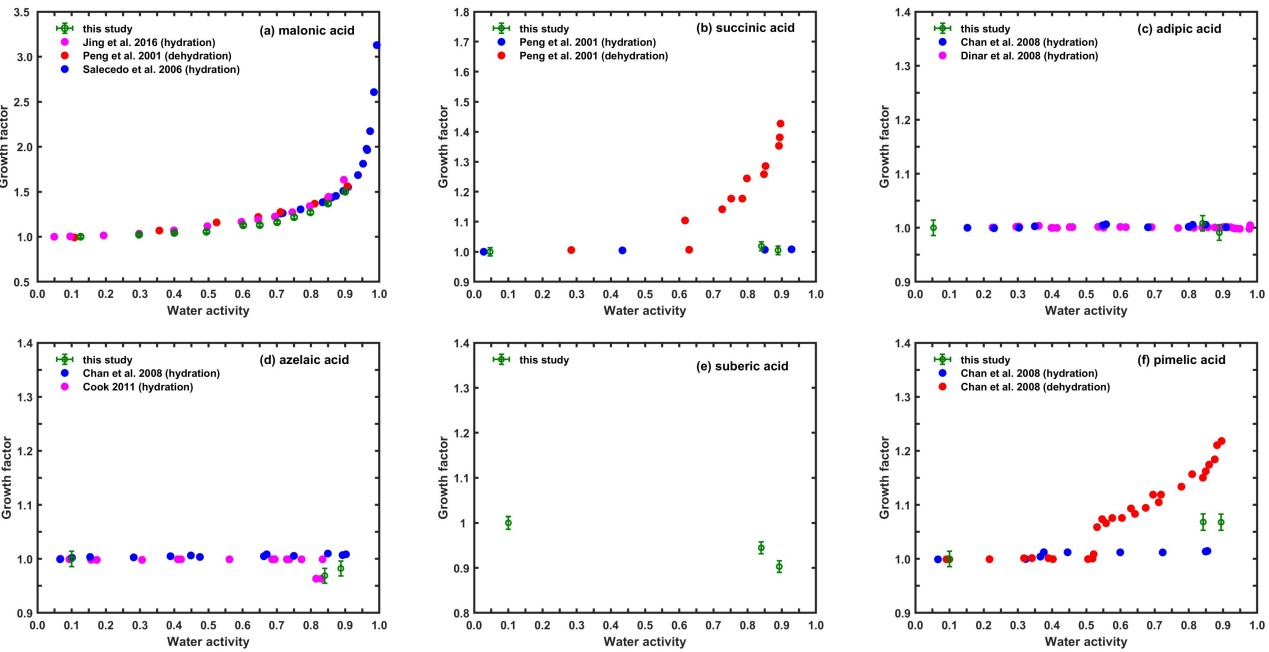

**Figure 1. The hygroscopic growth values of straight-chain dicarboxylic acids particles (200 nm) measured in this study and their individual comparison with previous studies.**

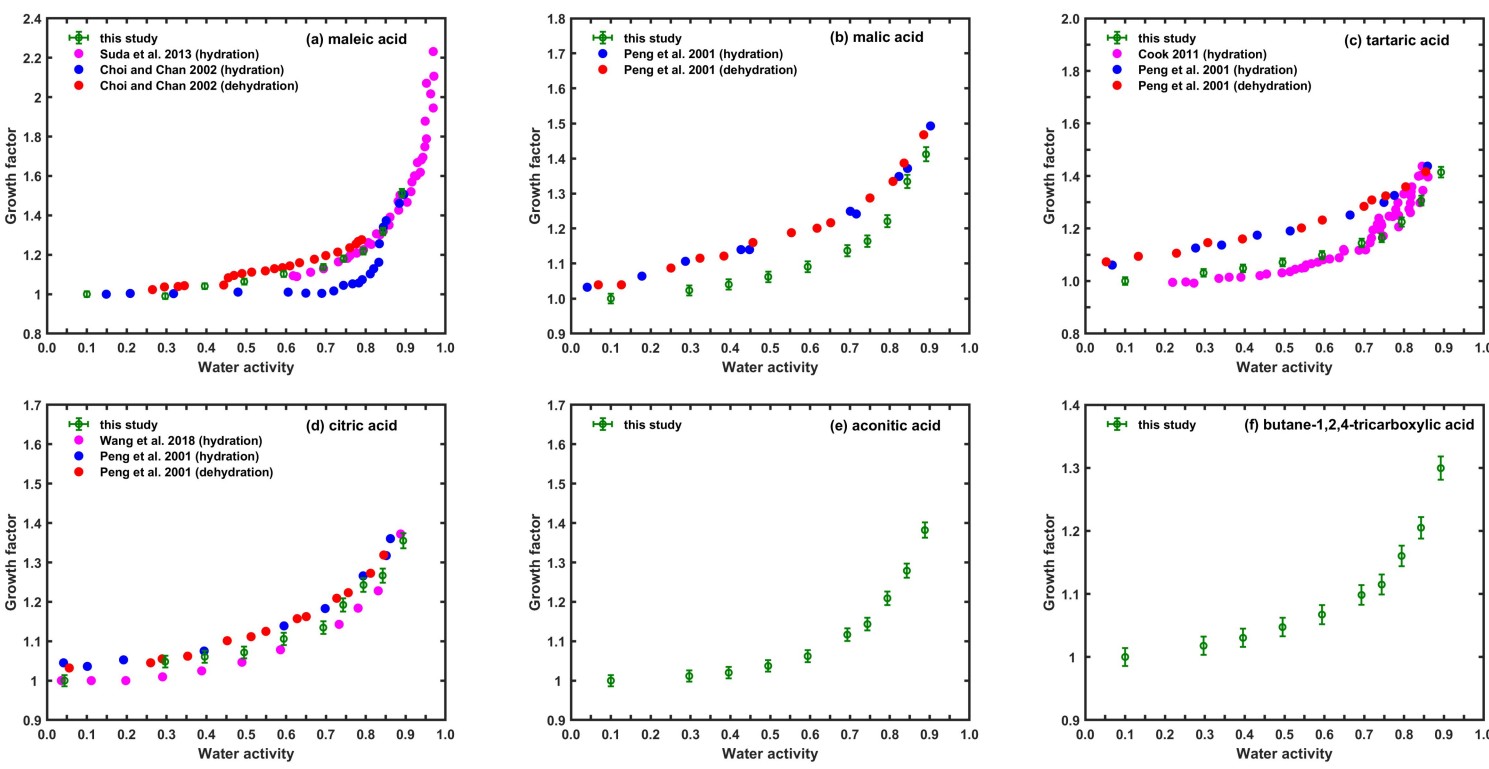

**Figure 2. The hygroscopic growth of straight-chain dicarboxylic acids with substitutions (a-c) and tricarboxylic acids (d-f) particles (200 nm) measured in this study and previous studies.**

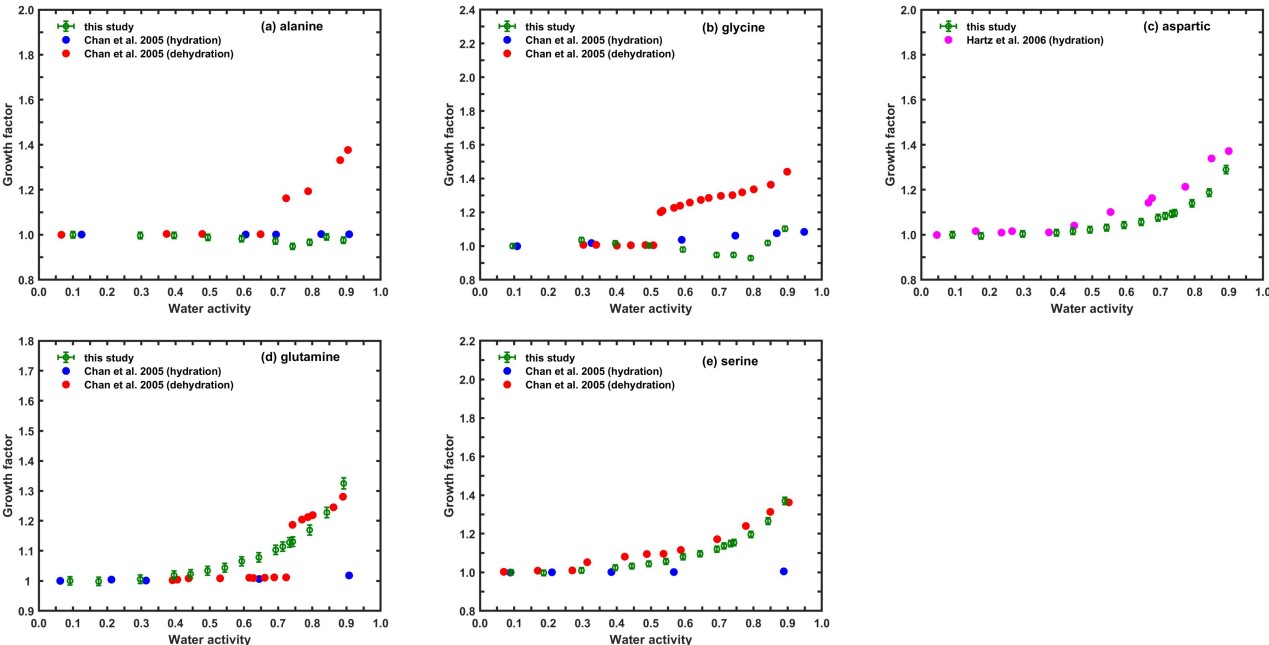

10          **Figure 3. Hygroscopic growth factors of amino acids, and comparison with previous studies.**

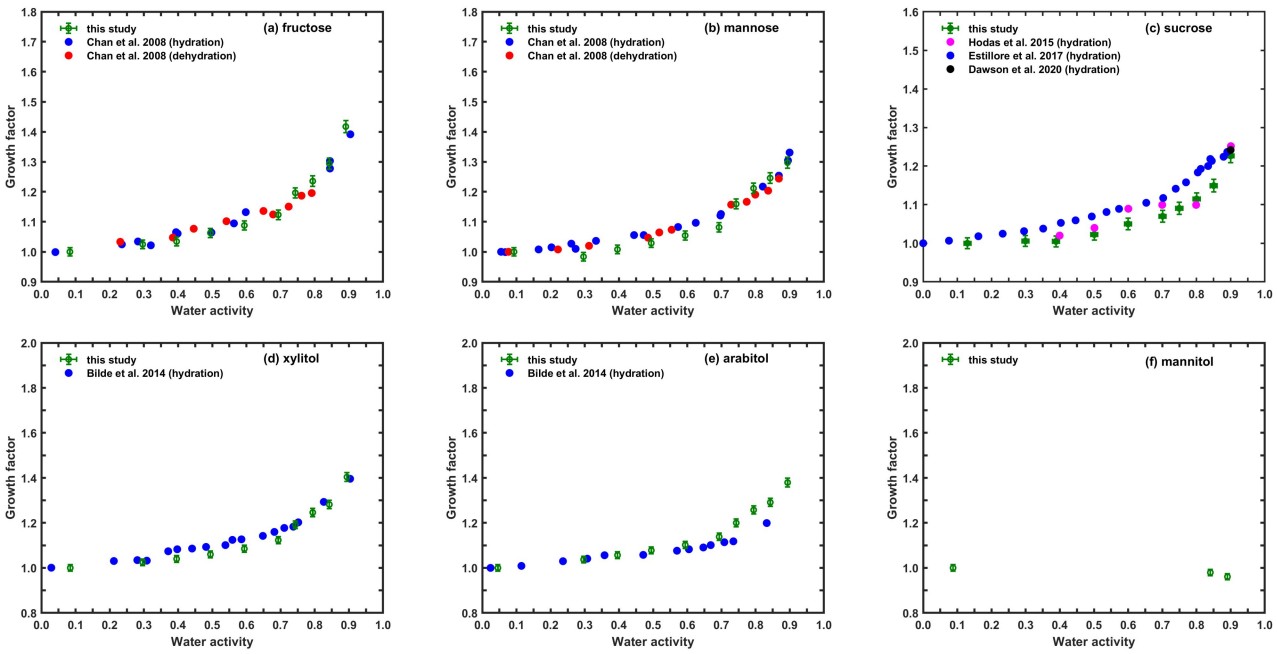

**Figure 4. Hygroscopic growth factors of sugars (a-c) and sugar alcohols (d-f), and comparison with previous studies.**

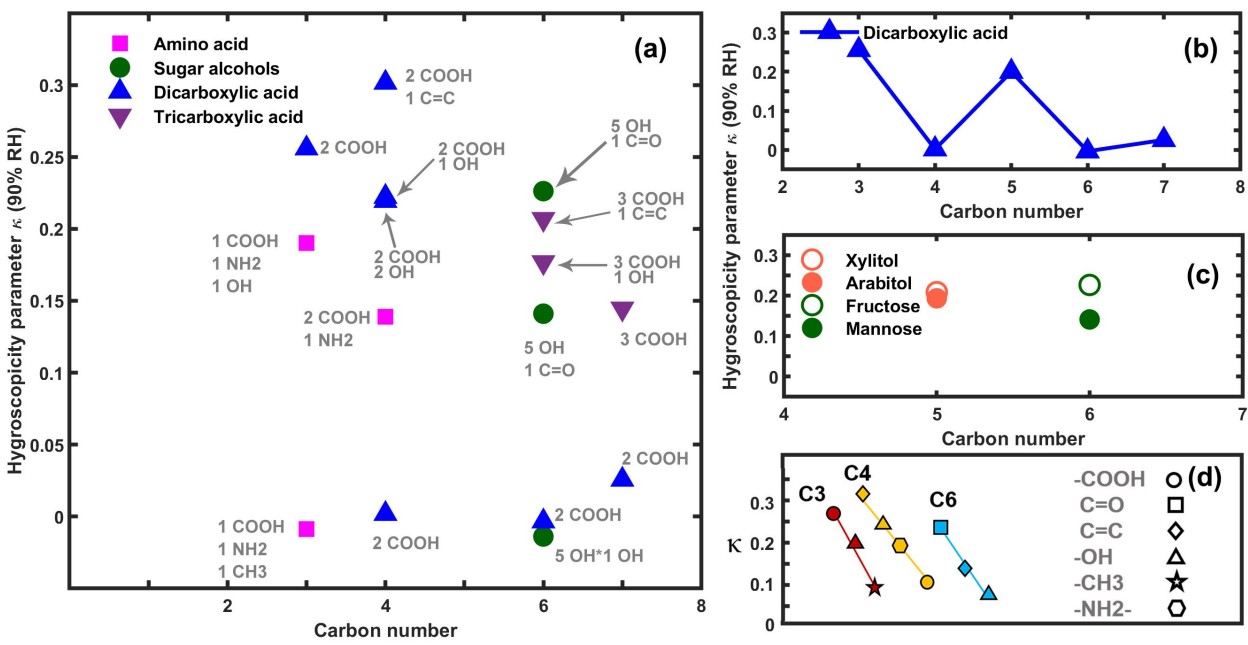

Figure 5. Hygroscopicity of organics as a function of carbon number (a); hygroscopicity of dicarboxylic acids vs carbon number (b); hygroscopicity of isomers (c); organic hygroscopicity as a function of their functionality with the same carbon number (d).

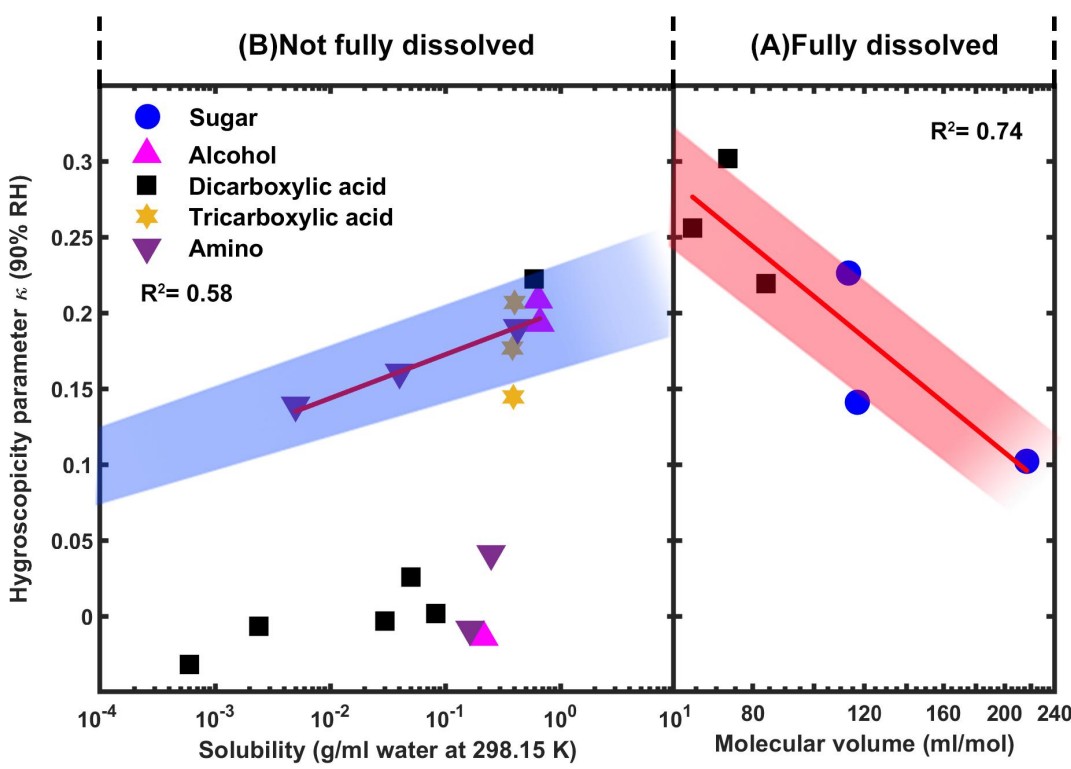

**Figure 6. Hygroscopicity of organic compounds as a function of molecular volume (A) and solubility (B).**

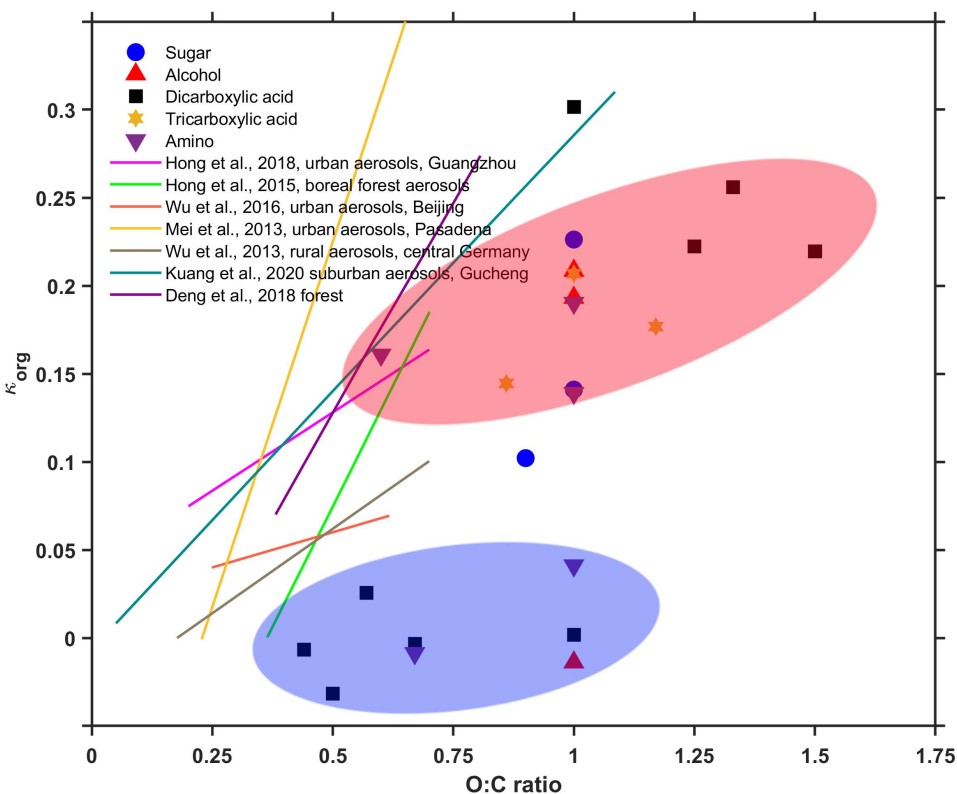

**Figure 7. Correlation between O:C ratio and $\kappa_{org}$, and comparison with previous literature results. Blue and red shades represent the fitting of results of non-hygroscopic and more hygroscopic organics, respectively.**