# Peer review of "Hygroscopicity of organic compounds as a function of organic functionality, water solubility, molecular weight and oxidation level"

_Atmospheric Chemistry and Physics, 2021_

## Author Comment (AC2)

**Answers to Referee #2**

The authors appreciate the critical and constructive comments that the reviewer has given, which will help us to improve the communication with other researchers for the understanding of HTDMA work. All the requested comments are replied and corresponding suggestions are addressed in the revised version of the manuscript.

Major comments:

The authors present hygroscopicity data for 23 organic compounds measured with an HTDMA instrument. They compare these data with two thermodynamic models (AIOMFAC and E-AIM) and conclude that the models do not represent the hygroscopicity well.

Unfortunately, the study is conceptually flawed. The experimental design allows studying the hygroscopic growth starting from dry conditions to elevated relative humidities. This way, as the authors illustrate in Fig. S2 for an inorganic salt (ammonium sulfate), they can only probe the properties of the aqueous system at relative humidities, beyond those corresponding to the solubility limit of each binary system. However, what is relevant for the atmospheric application, where an aerosol particle consist of a multitude of organic compounds, is the water activity (or hygroscopicity) of the liquid state, because the organics will not crystallize under realistic atmospheric conditions (e.g. Marcolli et al., 2004).

Reply: The reviewer thinks the organics will not crystallize under realistic atmospheric conditions and considers the experimental design, which we dried our aerosols and made them crystallize, is not atmospherically relevant and conceptually flawed.

However, some of the organics may crystallize under realistic atmospheric conditions. For instance, Chan et al. (2008) reported the efflorescence RH (ERH) is 55-59% for succinic acid, >85% for adipic acid, 51-53% for pimelic acid. Chan et al. (2005) reported the ERH for a series of amino acids is 67-70%, 53-55% and 72-74% for alanine, glycine and glutamine, respectively. These results suggested that these organic compounds will crystallize under relative high RHs, which is likely occur in the real atmosphere. Moreover, when organic aerosols mixed with inorganic components, crystallization was still observed in previous studies (Lei et al., 2014; Luo et al., 2020), though this process is composition-dependent. Furthermore, Pajunoja et al. (2015) used an aerosol bounce instrument (ABI) to directly characterize the phase state of different SOA generated in a flow tube reactor and found that most of the generated particles are solid or semisolid at intermediate RH, for example (<70% RH), which are under realistic atmospheric conditions (see a copy of their result in Fig. 1 below). These results further confirmed that organic compounds, even present in mixtures, could still crystallize under ambient conditions. In our study, we dried the aerosols to relatively quite low RH, ensuring the crystallization of the organic compounds that likely crystallize under real conditions. Thus, the measured hygroscopic growth at elevated RHs could represent their hygroscopic behavior under ambient conditions, forming a realistic dataset.

We agree that there remains some other organics that will not crystallize under realistic atmospheric conditions. Hygroscopicity of these organics could also be measured by current experimental design, as organic compounds that exist in liquid state will show continuous water uptake without prompt change in measured growth factor during hydration using HTMDA system. This was confirmed by our results of malonic acid and serine, which show continuous increase in growth factor upon wetting, indicating they are in liquid state. We assume these results are exactly what the reviewer expected.

Based on these aforementioned evidences, we conclude that to study the water uptake of particles from dry condition to elevated RH according to current experimental design is reasonable with highly atmospheric relevance. Moreover, it could also provide us some information that which compound at which RH conditions would crystallize. This will further benefit the deduction of the effect from mixing on the water uptake of mixtures, which might not be straight-forward and provide benchmark information for more complicated mixtures, for instance, ambient particles under various conditions.

[Figure]

Figure 1: A copy of Figure 1b of Pajunoja et al. (2015). Bounced fraction (BF) of aerosols as a function of relative humidity for flow tube generated aerosols using different precursors. When BF >0, aerosols are considered as solid or semisolid and aerosols at BF = 0 behave mechanically as liquids.

The HTDMA instrument as used by the authors probe only the thermodynamics of the 23 binary systems in the water activity range, which is accessible to bulk methods, the technique used here is simply not appropriate to determine thermodynamic data with high accuracy.

This is best illustrated by comparing the data presented in this work for the amino acid alanine with previous data by Chan et al. (2005): The authors observe no growth up to a relative humidity of 90 % which seems to be the highest they can reach in their setup. That is consistent with the data of Chan et al. upon humidification. But the data of interest are the ones when starting at high humidity and drying whith the metastable, subcooled binary liquid being probed. These can be compared to the thermodynamic data as done by Chan et al. (2005) for different UNIFAC parameterizations.

Reply: We understand the reviewer's worries, but we think the measurement for size-resolved aerosols, which is particularly relevant for atmospheric aerosols, is still a must. Thermodynamic data of bulk samples cannot always represent the ones at nano size ranges, particularly in the size range of the atmospheric aerosol accumulation mode (particle diameters around ~ $10^2$ nm). For instance, Lei et al. (2014) measured the growth factor of levoglucosan at 90% RH as 1.38 using a HTDMA system, while Chan et al. (2005) reported a value of 1.30 using an EDB. This clearly suggests that bulk data may not always be consistent with size-resolved ones. Cheng et al. (2015) also suggested that particle size can be an important factor influencing the solid-liquid equilibrium upon phase transition. Pöschl et al. (2015) and Reid et al. (2018) reported that thermodynamic properties of aerosols are strongly related to particle size, for instance, viscosity or diffusion coefficient. Therefore, different approaches with capabilities accessing different particle size ranges, requiring different sample volumes are essentially needed.

On the other hand, HTDMA has been a widely used technique in aerosol hygroscopicity studies since decades, including both field measurements and laboratory experiments. In our study, we calibrated our instrument using ammonium sulfate, a commonly-used reference in hygroscopicity community. Our obtained DRH and corresponding GF agree quite well with previous studies as well as the E-AIM prediction, confirming the accuracy of our measured data. Moreover, during our measurements, we observed a moderate water uptake for glutamine and serine particles, while no deliquescence was observed for both compounds by Chan et al. (2005) using an EDB, where bulk samples were collected. We believe that such a large discrepancy between our results and others cannot simply prove our method is not appropriate to determine thermodynamic data, but on the contrary, a strong evidence to confirm the difference between bulk samples and size-resolved particles. Hence, size-resolved data similar to ours is important, not only for filling the data gap due to the different capacities of techniques, but also helping the understand of the kinetic limitations for the mass transfer and transport of water molecules in the particle phase with the considering of particle size.

As the reviewer stated again that the data of interest are the ones when starting at high humidity and drying with the metastable, subcooled binary liquid being probed, we agree that these data are of interest. However, starting at dry conditions and measuring their hygroscopic behavior at elevated RH is also of interest, as some organic compounds may crystallize under relative high RHs, which is likely occur in the real atmosphere and how these crystallized organics behave towards changing ambient relative humidity is also very important for atmospheric science community. Moreover, the other reasons for its importance were also described in the answer to the first comment above clearly. We believe that a more thorough comparison of the results during a complete water cycle, including both hydration and dehydration is necessary for future studies.

In addition to this conceptual flaw, the experimental data raise questions as well. In Fig. 1(a) the authors observe considerable deviations between their data for malonic acid and the AIOMFAC model in the range of 60% to 80 % RH. However, if I compare AIOMFAC with existing experimental bulk data the agreement is very satisfactory in the same range of humidities:

As the bulk data are clearly the reference, the authors need to explain the difference to those, before concluding that the thermodynamic models are failing for the binary aqueous components. One possible reason could be that the authors are using ideal mixing to convert from size growth to mass concentration?

Reply: We are quite confused with current comment raised by the reviewer. In our study, we did not use AIOMFAC, but E-AIM. Moreover, in our original manuscript, we stated that "Figure 1a also shows that the hygroscopic behaviour of Malonic acid was well represented by the E-AIM but overestimated by the UManSysProp", which agrees with the results the reviewer presented as well. We agree that there are some slight deviations between the measurement and the model predictions in the RH rang of 60% to 80%. But this is also similar to the figure the reviewer presented to us at $a_w$ of 60%-80% RH, see figure below. Based on current comments, we repeated the measurements again for malonic acid, presented below, and confirmed that the hygroscopic behaviour of malonic acid was well represented by the E-AIM, considering the measurement uncertainties.

[Figure]

Figure 1: Measured and predicted growth factor as a function of relative humidity (RH) for malonic acid as.

Moreover, the bulk data may not always represent the reference. For instance, Chan et al. (2008) measured the hygroscopicity for a series of monosaccharides and disaccharides using an EDB. Though the predictions agree well with the measurements for monosaccharides, there remains significant deviations between model predictions and measurements for disaccharides. Similarly, in our study, though the hygroscopic growth of malonic acid was well represented by the E-AIM, some deviations between model predictions and measurements were still observed for other compounds, for instance, amino acids. The potential sources for the discrepancies between the model predictions and the measurements, including both bulk measurements and size-resolved ones could all be due to that the phase transition, intermolecular interactions as well as other non-ideality were not considered in the simulations. Detailed discussions for phase transition and intermolecular interactions were given in our manuscript.

Considering the last suggestion the reviewer raised, we agree that to calculate size growth from mass concentration assuming ideal mixing may contribute to the deviations between model and measurements. Clearly, in E-AIM, GF values of the pure component was originally calculated according to Eq. (1):

$$GF = \left[ 1 + \frac{n_w M_w / \rho_w}{\sum_i (m_i / \rho_i)} \right]^{\frac{1}{3}}$$

where $M_w$ is the molar mass of water, $\rho_w$ is the density of water, $m_i$ and $\rho_i$ are the mass and density of the pure component i, $n_w$ is the mole number of water at a certain RH. We agree with the reviewer that the GF value from E-AIM were calculated from mass concentration to size growth assuming ideal mixing. As suggested by the reviewer that the non-ideality of the aqueous solution should be considered, we made a sensitivity analysis for ammonium sulfate due to the limited data for the density of the aqueous solution of our studied compounds at various RH conditions. Specifically, we calculated the GF for ammonium sulfate by using the density of their aqueous solution ($\rho_{solution}$) as Eq. (2):

$$GF = \left[ \frac{(\sum_i m_i + n_w M_w) / \rho_{solution}}{\sum_i (m_i / \rho_i)} \right]^{\frac{1}{3}}$$

After the deliquescence of ammonium sulfate, the difference in GF value using different density data is less than 0.01, well within the measurement uncertainties. For other organic compounds, the difference between the density of their aqueous solution with water is relatively smaller than that of ammonium sulfate. Hence, we estimate the difference in GF value might be even smaller for other organic compounds by using different density values. Therefore, we think to calculate size growth from mass concentration assuming ideal mixing might not be the major reason for the discrepancies between the model predictions and the measurements.

And a last comment. Sucrose has been used as aproxy for viscous or even galssy aerosol particles during the last years in a large number of studies. Its hygroscopicity is very well etsablished. However, the authors do not at all comapre their data with data availbel in the literature. Besides this problem, it is obvious that they are not aware of the kinetic limitations to water uptake by viscous aerosol. Most likely, the short residence time in the HTDMA limits the water uptake for aqueous sucrose at intermediate and low humidities. This can be seen by comparing the data of this work with standard paramterizations for hygrscopicity of sucrose (e.g. Zobrist etal, 2011):

Reply: We agree with the reviewer that comparison with the other data for sucrose should be conducted. We therefore compared our results with the others using similar HTDMA system but with different residence time. We noticed that at intermediate and low humidities, our measured values for sucrose were lower than the other two. This could be due to the short residence time of our HTMDA system, limiting the water uptake for aqueous sucrose, as suggested by the reviewer.

[Figure]

Figure 2: Comparison of the measured growth factor for sucrose as different RH with other studies using different HTDMA.

For this reason, we added a new section in atmospheric implication to discuss the effect of residence time of humification on the water uptake of particles. Besides sucrose, we also made a thorough comparison of the hygroscopicity for other compounds using different techniques. Based on the comparison results, the influence of the residence time on the organic hygroscopicity is difficult to conclude, implying a more complicated mechanism for the discrepancies between different studies using different techniques. On the other hand, we also found that our measured GF value at high RH, e.g., 90% RH was quite close to other results, suggesting equilibrium might already been reached. In our manuscript, there is an important section discussing the role of different physico-chemical properties on the organic hygroscopicity. The analysis in this section was all based on the measured GF at the highest RH we reached in our system. We hope at this point (i.e., 90% RH), the influence from residence time, which may affect the equilibrium state of particles upon hydration, can be largely reduced and we suggest a comprehensive study to thoroughly investigate at which condition the liquid-solid equilibrium of organics could be reached upon wetting is needed. Particularly, the new section was listed below:

[revised manuscript text omitted]

---

## Author Response (AR1)

Dear Editor,

We would like to thank all the reviewers for their detailed and constructive comments and suggestions. Our responses addressing reviewers' comments point-by-point are given below. The revised manuscript tracking the changes is also attached in the end of this file.

**Best Regards**

Shuang Han and Juan Hong

**Answers to Referee #1**

The authors appreciate the detailed and constructive comments that the reviewer has given, which will help us to improve the structure and content of our work. All the requested comments and suggestions are addressed and introduced to the revised version of the manuscript.

Major comments:

The authors fall short of establishing/communicating the work's novelty. Several findings have been explored in past works (functional groups, molar mass, O:C, solubility and deliquescence). The authors have done a good job finding these studies. However, authors should discuss and emphasize their own contributions.

Reply: Thanks for the reviewer's specific comments. We revised our abstract, see the reply to the second comment below and the original introduction part and added new material discussing the novelty of our work. The revised introduction is listed below with modification highlighted in yellow:

[revised manuscript text omitted]

Further comments on the abstract: As stated above, findings summarized here should emphasize the novelty of the current work. The abstract should end with implications.

Reply: We revised our abstract as: "Aerosol hygroscopicity strongly influences the number size distribution, phase state, optical properties as well as multiphase chemistry of aerosol particles. Due to the big number of organic species in atmospheric aerosols, the determination of the hygroscopicity of ambient aerosols remains challenging. In this study, we measured the hygroscopic properties of 23 organics including carboxylic acids, amino acids, sugars and alcohols using a Hygroscopicity Tandem Differential Mobility Analyzer (HTDMA). Earlier studies have characterized the hygroscopicity either for a limited number of organic compounds using similar techniques or for particles at sizes beyond the micro-scale range or even bulk samples by other methodologies. Here, we validate these studies and extend the data by measuring the hygroscopicity of a broader suite of organics for particles with size under the submicron range that are more atmospheric relevant. Moreover, we systematically evaluate the roles of related physico-chemical properties that play in organic hygroscopicity. We show that hygroscopicity of organics varies widely with functional groups and organics with the same carbon number but more functional groups show higher hygroscopicity. However, some isomers, which are very similar in molecular structures, show guite different hygroscopicity, demonstrating that other physico-chemical properties, such as water solubility may contribute to their hygroscopicity as well. If the organics are fully dissolved in water (solubility >  $7 \times 10^{-1}$  g/ml), we found that their hygroscopicity is mainly controlled by their molecular weight. For the organics that are not fully dissolved in water (slightly soluble: 5×10-  $^{4}$  g/ml < solubility < 7×10-1 g/ml), we observed that some of them show no obvious water uptake, which probably due to that they may not deliguesce under our studied conditions up to 90 % RH. The other type of slightly soluble organics is moderately hygroscopic and the larger their solubility the higher their hygroscopicity. Moreover, the hygroscopicity of organics generally increased with O:C ratios, although this relationship is not linear. Hygroscopicity of organic compounds were also predicted by two thermodynamic models, including the Extended Aerosol Inorganics Model (E-AIM), and the University of Manchester System Properties (UManSysProp). We show that hygroscopicity results of almost all organic compounds except those tricarboxylic acids were poorly represented by the UManSysProp and the E-AIM over-estimated the hygroscopicity of all amino acids. These discrepancies were likely due to that both models do not consider phase transition and intermolecular interactions of these selected compounds in the simulations. These results may further improve our understandings of the interactions between organics and water molecular and will benefit the estimate of the hygroscopicity and CCN (cloud condensation nuclei) activities of any mixtures, for instance, ambient mixtures based on known composition data.

17 – with additional functional groups – addition to what? This is more complicated than just the addition of functional groups. Carbon number matters as well.

Reply: We agree that carbon number also matters and the comparison of hygroscopicity among different organics would be quite difficult that many other properties rather than functional groups may also be important. Here, in our statement, we actually meant to compare the hygroscopicity of the organic compounds with the same carbon number. We rephrased our statement in the abstract as: "<mark>We show that hygroscopicity of organics varies</mark> widely with functional groups and organics with the same carbon number but more functional groups show higher hygroscopicity."

**18 - It sounds like you mean isomers. This statement is ambiguous**

Reply: We rephrased the sentence in the abstract as: "However, some isomers, which are very similar in molecular structures show quite different hygroscopicity, demonstrating that other physico-chemical properties, such as water solubility may contribute to their hygroscopicity as well."

**23 – "moderately"**

**Reply: We rephrased the word "moderate" as "moderately" accordingly.**

Uncertainty estimates are needed. As it stands, scatter in the data is used to discuss morphology. Although this is a nice discussion, some error bars and acknowledgement of the limitations of the measurement would lend more credibility to these claims.

Reply: Yes, we agree. We added a part introducing how we estimated the measurement uncertainties for the measured GF in the Measurement section (Sect. 2). The estimated uncertainties were added for each measured dot in Figure 1, 2 and 3 in the revised manuscript with according discussions. The measurement uncertainty was introduced before the third paragraph in Sect. 2 as:

"Swietlicki et al. (2008) summarized the potential sources of error in HTDMA measurements and concluded that the reliability of the measured data is strongly associated with the stability and accuracy of DMA2 RH as well as the accurate measurement of particle diameter by DMAs. According to Mochida and Kawamura (2004), the uncertainty in the measured GF can be calculated by Eq. (2):

$$\sqrt{\left(GF\frac{\sqrt{2}\varepsilon_{Dp}}{Dp}\right)^2 + \left(\varepsilon_{RH}\frac{dGF}{dRH}\right)^2},\tag{2}$$

where GF is the measured growth factor with respect to any measured RH,  $\varepsilon_{Dp}$  and  $\varepsilon_{RH}$  are the errors in the measured Dp and RH. In our system, the accuracy of DMA2 RH was maintained to be ±1% and the uncertainty for the mobility diameter was ±1% according to PSL (Polystyrene Latex particles) calibration. Hence, for our system,  $\varepsilon_{RH}$  and  $\varepsilon_{Dp}$ /Dp are 1% and 0.01, respectively. The calculated uncertainty according to the above-mentioned method is added in the measured GF in the following section."

An example of the revised figure for the measured GF as a function of water activity (response for the third minor comment) for carboxylic acids including measurement uncertainties is shown below. Revision of the other part of our manuscript was made accordingly. Here, the revised discussion (the last two paragraphs of Sect. 4.1.1) for current figure was also shown as an example:

"The humidograms of the three dicarboxylic acids with substitutions (i.e., double bond or hydroxy group) are illustrated in Fig. 1b. The continuous water uptake indicates that these particles may be at liquid state under dry conditions. We observed a small leap of the GFs

from 80 % to 85 % RH, implying that these particles were only partially deliquesced and further dissolution occurred at elevated RH. However, considering the measurement uncertainties, the statement of the partial deliquescence could not be fully confirmed and thus further evidence from other measurements is needed. On the other hand, the E-AIM could well represent their hygroscopic properties at RH between 10 -90%, while there are still some deviations between the measurements and the UManSysProp predictions, especially for malic and tartaric acids, even taking into account of their measurement uncertainties. This could be due to that the UManSysProp is a more simplified model taking into account less input data or parameterization. For instance, the phase state or the dissociation process of the studied compound at different conditions could not be assumed or considered in the simulation. Therefore, additional processes or properties should be included into this model for the further improvements of its predictions.

A similar gradual phase transition was observed for aconitic acid and citric acid (Fig. 1c), while the other tricarboxylic acid showed continuous hygroscopic growth over the studied RH range, indicating no obvious phase change for these particles upon hydration. However, it is quite interesting to note that predictions from the UManSysProp become more approaching to the experimental data, especially above 80 % RH, considering the measurement uncertainties."

Figure 1: Hygroscopic growth curves of straight-chain dicarboxylic acids (a), dicarboxylic acids with substitutions (b) and tricarboxylic acids (c) particles (200 nm). Points represent the measurement data; the solid lines indicate the E-AIM predictions (solid, non-hygroscopic organic GF=1) and the dashed lines show the UManSysProp calculated predictions.

And the original line 140: "Considering the measurement uncertainties, no water uptake is observed for alanine particles, which has also been reported in previous works (Chan et al., 2005; Darr et al., 2018)."

Some of the conclusions are not supported by the data. On line 184, the authors discuss the order in which the functional groups contribute to hygroscopicity. Is this statement quantitative? If so, what is the observed partial derivative of kappa with respect to each functional group? The statement seems to have little connection to the data presented in Figure 4.

Reply: Unfortunately, this statement is only a qualitative description based on currents results in our study instead of a quantitative comparison. We agree such conclusion in our manuscript might be too strong and we revised the text in the original line 184-185 as: "By summarizing the results in current study,  $\kappa$  increased with the functionality in the following order: (-CH3 or -NH2) < (-OH) < (-COOH or C=C or C=O). However, it has to be noted that this comparison is quite qualitative, might be ambiguous and further evidence from other organic compounds is needed in order to drive a more general conclusion."

On the other hand, our Fig. 4a in the manuscript is an overview of the measured hygroscopicity of the 23 organics with different functional groups. Fig. 4b is described by line 189-192 in the original manuscript. Fig. 4c is explained in line 192-194, while Fig. 4d is actually a schematic illustration for the content in line 178-185, which tells how the hygroscopicity of organics with the same carbon backbone number but different functional groups varies.

Restructuring of particles was observed, and this resulted in a negative growth factor. This was shown and discussed in the main text and in Figure S3, which shows severe discontinuities in water uptake for amino acids. Some of the restructuring-sizing error could be avoided by sizing the particles wet, following Nakao et al. (2014). This should be discussed. Nakao et al. (2014), Droplet activation of wet particles: development of the Wet CCN approach, Atmos. Meas. Tech., 7, 2227–2241.

Reply: We carefully read the paper by Nakao et al. (2014). It is a quite interesting paper. Here, in current work, proper discussion citing Nakao et al. (2014) was added in the original line 145: "The sizing of these structurally-rearranged particles, especially at lower RH range, will be erroneous as the volume change of the particles upon wetting may not only due to the water absorption but also the compaction of the original particles. This phenomena complicates the accurate estimation of the actual water amount absorbed by the particles due to their intrinsic hygroscopicity. In a recent study by Nakao et al. (2014), in order to avoid the influence from particle restructuring upon wetting, they sized wet particles without drying after generation and studied their droplet activation using a wet CCN. This approach they introduced might be an easier attempt, offering an unique solution for current problem from particle restructuring during the hydration processes."

Regarding restructuring, the residence time of the HTDMA is mentioned (2.7 seconds) but the authors do not bring this into the discussion. The authors should mention how this 2.7 s residence time affects particle restructuring, and how this instrument compares to other works.

Reply: Yes, we did a thorough literature review for the hygroscopicity measurements using other techniques, carefully compared their results with ours and brought this content into the discussion.

We extended the discussion in Section 5: Atmospheric implication including the effect from residence time as: "Previous studies (Swietlicki et al; 2008; Duplissy et al., 2009; Wu et al., 2011; Suda et al., 2013) suggested that the residence time for humidification may also potentially influence the observed water uptake of particles as the measured particles, especially for some organic compounds, may not reach their equilibrium humidified sizes during a quite short time of wetting. However, extending the humidification time for hours using the EDB, no water uptake was also observed for most of our studied dicarboxylic acids

(Chan et al., 2008) as well as for glycine and alanine particles (Chan et al., 2005), which was also confirmed by Darr et al., (2018), using another different measurement technique, i.e., ATR-FTIR (Attenuated Total Reflection Fourier Transform Infrared) with a residence time of 2 minutes. Estillore et al. (2017) reported a quite similar value of GF at 90% RH (1.24) for sucrose as ours using a different HTDMA with a much longer residence time (40s), similar as the one by Hodas et al. (2015) based on DASH-SP (Differential Aerosol Sizing and Hygroscopicity Spectrometer Probe) with a residence time of 4s. However, at intermediate RHs, our measured GF were much lower than theirs. Moreover, for glutamine and serine particles, no deliquescence was observed by Chan et al. (2005) even with a much longer residence time, while in our study a moderate water uptake for both compounds were observed. Using an STXM (Scanning Transmission X-ray Microscopy) with a residence time of 5-10 minutes, Piens et al. (2016) obtained a lower GF of fructose compared to ours, which should not be caused by the evaporation losses due to its low volatility. These aforementioned comparisons pointed out the influence of residence time on the observed water uptake of particles might not be conclusive. Therefore, other technical reasons should be raised for the measurement discrepancies between different instruments and studies using similar technique but different residence times should be suggested for understanding the effect from residence time on hygroscopicity."

There is a severe disagreement between the model and the measurements for some compounds, but the authors do not attempt to improve the prediction of water uptake by any calculation or modification to the models.

Reply: In this work, we aim to provide unambiguous HTDMA data of a large suite of organics and understand the roles of different physico-chemical properties that play in organic hygroscopicity, which may benefit the improvement of predictions of different thermodynamic models. These data may help the model developer to further understand the limitations and applicability of current models. However, to make obvious improvement of these models, we may need more information, such as a sufficient acquaintance of the simulation procedure and involving parameterization, which may not be able to be resolved in current study.

We appreciate this comment suggested by the reviewer and made some limited attempts that we can only access to improve the model predictions. Specifically, we changed some of the input properties of the studied organics in both models. For instance, we changed the phase state of the studied organic compound occurring in the liquid droplet. However, no significant change was observed in the obtained GF. Moreover, for the amino acids we studied, we used a reduced surface tension in both models instead of that of water due to their surface partitioning. However, the discrepancies between the measurements and the predictions became even larger. These are so far the attempts we made to improve the predictions, though it didn't work out. Future work with substantial efforts, requiring detailed discussions with the model developer are needed as these models we used are online versions that only limited modification can be made.

The authors do not discuss the disagreement between UManSysProp and E-AIM. Why do these models behave differently? There is valuable information here and it should be discussed.

Reply: Both models are online predictors for the water uptake of some specific compounds or their mixtures. There are some differences between these two models. For instance, E-AIM

mainly calculates the properties of bulk samples, while size-resolved information can be obtained from UManSysProp. Moreover, the phase state of the studied compound exist in the particles or droplets could be assumed in E-AIM, while no such input data can be included in UManSysProp, which is more simplified model that only molecular structure, surface tension and size can be modified. From the assumptions and input parameterization used for both models, we can deduce that the calculation mechanism or the involving physico-chemical processes might be different in these two models. This could also be the major reason for the disagreement between E-AIM and UManSysProp. We brought this information into our discussion (the third paragraph in Sect. 4.1.1) as: "This could be due to that the UManSysProp is a more simplified model taking into account less input data or parameterization. For instance, the phase state or the dissociation process of the studied compound at different conditions could not be assumed or considered in the simulation. Therefore, additional processes or properties should be included into this model for the further improvements of its predictions."

**Minor Comments**

Solubility: Line 219, 241, and elsewhere – there are too many "types" here, consider clarifying these paragraphs by using more specific and consistent terminology for solubility regimes. See, for example, Petters and Kreidenweis (2008). Petters and Kreidenweis (2008), A single parameter representation of hygroscopic growth and cloud condensation nucleus activity – Part 2: Including solubility, Atmos. Chem. Phys. 8, 6273–6279.

Reply: Yes, we clarified these paragraphs accordingly.

Line 206-207 was revised the second sentence in the first paragraph in Sect. 4.2.2 as: "(B) compounds that are not fully dissolved (slightly or sparingly soluble compounds with solubility in the range between 1e-3 to 3e-1 g/ml or saturated regime) in the aqueous droplets under 90 % RH condition."

Line 219-220 was revised the fifth sentence of the second paragraph in Sect. 4.2.2 as: "Compared to these non-hygroscopic slightly/sparingly soluble organic compounds, there are some other slightly/sparingly soluble organics, showing moderately water uptake with  $\kappa$ values larger than 0.1."

Line 241 was revised the fourth sentence of the second paragraph in Sect. 4.2.3 as "The other slightly/sparingly soluble organics shaded in red area in Fig. 6 is a moderate-hygroscopic group with a slightly stronger O:C-dependence."

Implementation of Kohler theory: On line 87, in the equation, ((1 - Ke/RH)Ke)/RH could be simplified to (Ke/RH-1).

Reply: Thank for your suggestion. Considering your suggestion, we simplified the equation as following:

$$\kappa = (GF^3 - 1)(\frac{Ke}{RH} - 1)$$

Because growth factor as a function of RH is diameter dependent, the results would be more general if the RH is divided by the Kelvin term, here expressed Ke (on line 87). This means that instead of RH you have growth factor as a function of water activity. aw = RH/Ke.

Reply: Thank you for your specific comments. We agree with your suggestion and revised the equation in Sect. 2 as following:

$$Ke = exp\left(\frac{4\sigma_{sol}M_{w}}{RT\rho_{w}D(RH)}\right),$$
$$RH/100\% = a_{w}Ke,$$
$$(GF^{3}-1)(1-a_{w})$$

aw,

where  $a_w$  is the water activity,  $M_w$  and  $\rho_w$  are the molar mass and the density of pure water at temperature *T*, respectively;  $\sigma_{sol}$  is the solution droplet surface tension, which was assumed to be the surface tension of water (0.072 J m-2) and *R* is the ideal gas constant.

Also, the figures (Figure 1, Figure 2 and Figure 3) were plotted against water activity instead of RH in the revised manuscript.

33 – the citation does not match the statement in any way

Reply: We replaced "Wang et al., 2015" with "Seinfeld and Pandis, 2016".

147,147 – how do you know that the water mass fraction always increased when the growth factor "shrank or grew"? This calculation should be described here or in the supplement.

Reply: We originally meant that according to EDB measurements, the water mass fraction increased although our measured GF shrank. We deleted current sentence in our revised manuscript.

Figures: In general, the thick grey lines behind the data are distracting (all figures)

Reply: Yes, all figures without grids were used instead in the revised manuscript.

Figure 4 – Size and style of panel D should be consistent with the rest of the figure

Reply: Yes, the figures were modified accordingly in the revised manuscript.

Figure 6 – fonts are difficult to read and should be enlarged. There should be spaces between the words.

Reply: Yes, the figures were modified accordingly.

The table of contents in the supplement is very confusing. Why are figures each described twice, except for Figure S4? Also: there is no Figure S4. Please clarify this text.

Reply: We agree that the table of content is quite confusing and we deleted the table of content in the supplement.

**Technical corrections**

The references arranged either alphabetically or chronologically (forward or reverse).

Reply: Yes, we rearranged the references accordingly.

Do not capitalize the names of organic acids (line 114 and elsewhere).

Reply: Thank you for your comments, we revised the names of organic acids in the manuscript.

Caption of Figure 1 - (a) and (b) are listed together but the text differentiates these. Please be more descriptive in the caption as well

Reply: We revised the figure caption in Fig. 1 as: "Figure 1: Hygroscopic growth curves of straight-chain dicarboxylic acids (a), dicarboxylic acids with substitutions (b) and tricarboxylic acids (c) particles (200 nm). Points represent the measurement data; the solid lines indicate the E-AIM predictions (solid, non-hygroscopic organic GF=1) and the dashed lines show the UManSysProp calculated predictions."

Caption of Figure 1 – space between 200 and nm

Reply: We added a space between 200 and nm.

31 – McFiggans capitalize F

Reply: We capitalized F in "McFiggans".

37 – "large" not big

Reply: We switched the word from "big" to "large".

44 – "relies"

Reply: Thank you for your comment, we changed "rely" to "relies".

47 – "(E-AIM)," add comma, define UManSysProp

Reply: Yes, we agree with your comments. We defined UManSysProp in line 47 as well as in the abstract, and we added comma after (E-AIM).

55 – "molecular interactions" ... "and water."

Reply: In the revised manuscript, current sentence was deleted.

62 – "experimental hygroscopicity data for organics"

Reply: We added "experimental" into the revised sentence.

66 - "experimentally-determined"

Reply: Due to a revision of the introduction part of our manuscript, current statement was deleted.

68,76 –Strike "self-assembled" or replace with, e.g., "home-built", "custom-made", "custom engineered", or "HTMDA built in-house." "Self-assembled" implies that the instrument assembled itself spontaneously.

Reply: Yes, we agree with your comments, we changed "self-assembled" to "custom-made".

74 – "using ultrapure"

Reply: We added "using" before "ultrapure".

Table 1 title: "Substances"

Reply: We revised "Substance" to "Substances".

Table 1 header row: "Supplier, purity"

Reply: We changed "Supplier / purity" to "Supplier, purity".

75 – physicochemical –either "physico-chemical" or "physicochemical" – make consistent throughout paper

Reply: Yes, we used "physico-chemical" throughout the whole paper.

79 – under dry conditions

Reply: Thank you for your advice, and we revised "dry condition" to "under dry condition"

80 - "the detailed schematic"

Reply: We changed "the detailed principle" to "the detailed schematic".

81 – period after (2013)

Reply: We added "period" after "(2013)".

88 - Mw is erroneously included inside the subscript of sigma

Reply: We corrected this equation:  $Ke = exp \left(\frac{4\sigma_{sol}M_{W}}{RT\rho_{W}D(RH)}\right)$ .

89, 90 – italicize T and R

Reply: Yes, we used the italicized version.

94 – UManSysProp is defined here – should be defined above, or in both places

Reply: We agree, and we defined UManSysProp in the abstract.

109 - please clarify sentence

Reply: We revised the third sentence of the first paragraph in Sect. 4.1.1 as: "To achieve a comprehensive overview of the hygroscopicity of carboxylic acids, we measured the water uptake of several common straight-chain dicarboxylic acids in the atmosphere and further extended the hygroscopic measurements for dicarboxylic acids with substitutions and tricarboxylic acids."

**115 – "dicarboxylic acids"**

Reply: We modified "dicarboxylic acid" to "dicarboxylic acids".

128 – period after "RH."

Reply: We revised this sentence.

130 – "relatively higher"

Reply: Due to a revision of the results part of our manuscript, current statement was deleted.

131,132 – "current models have insufficient data" – break sentence into more than once sentence

Reply: The sixth sentence of the third paragraph in Sect. 4.1.1 was revised as: "This could be due to that the UManSysProp is a more simplified model taking into account less input data or parameterization. For instance, the phase state or the dissociation process of the studied compound at different conditions could not be assumed or considered in the simulation. Therefore, additional processes or properties should be included into this model for the further improvements of its predictions."

**134 - visible how? Detectable?**

Reply: The last paragraph of Sect. 4.1.1 was revised as: "A similar gradual phase transition was observed for aconitic acid and citric acid (Fig. 1c), while the other tricarboxylic acid showed continuous hygroscopic growth over the studied RH range, indicating no obvious phase change for these particles upon hydration. However, it is quite interesting to note that predictions from the UManSysProp become more approaching to the experimental data, especially above 80 % RH, considering the measurement uncertainties."

136 – "lower RH."

Reply: We revised our manuscript, current statement was deleted.

138,139 – "Continuous water uptake"

Reply: Thank you for your comments, we deleted "The", corrected to "Continuous water uptake".

142,143 - "Previous studies"

Reply: We revised "Previous literatures" to "Previous studies".

**144 – remove period after "2018)"**

Reply: Yes, we removed "period" after "2018)".

147 – remove comma after "though"

Reply: We deleted current phrases in the revised manuscript.

147 – "shrank or grew slightly"

Reply: We deleted current phrases in the revised manuscript.

148 – agrees with which previous results? Both of the prior citations? Please elaborate.

Reply: The whole section was revised and detailed literature comparison was given in the second paragraph in Section 5.

148,149 – "Actually, it cannot be defined" – this sentence is unclear, please rephrase. Remove or replace the word "actually."

Reply: We removed the sentence.

150 - "are generally in better agreement with"

Reply: We revised to "are generally in better agreement with".

155 – "growth was" Reply: We corrected "growths were" to "growth was".

160 - "literature"

Reply: We changed "literatures" to "literature".

**163 - please rephrase**

Reply: We rephrased the second paragraph in Sect. 4.1.3 as: "Similarly, phase transition or microstructural rearrangements of particles was not included in the models. Thus, these sugars and sugar alcohols were generally less hygroscopic than the values predicted by the E-AIM (except L-arabitol) under low RH conditions. However, at elevated RHs, whereas particles are fully dissolved, the E-AIM predictions agree well with most of the measured hygroscopic GFs within the measurement uncertainties."

Figure 2 (Figure 3 in the manuscript): Hygroscopic growth curves of sugars (a) and alcohols (b). Points represent the measurement values; the solid lines indicate the E-AIM predictions (solid, non-hygroscopic organic GF=1) and the dashed lines represent the UManSysProp calculated predictions.

168 – "Note that" Reply: Yes, we corrected to "Note that".

177 – "adding an"

Reply: Thank you for your comments, we revised this sentence to "adding an carboxylic acid group...".

209,210, ... - here and elsewhere, capitalize Hoff

Reply: We capitalized "Hoff" throughout the whole paper.

220 - "moderately"

Reply: We corrected "moderate" to "moderately".

221 - "hygroscopicity"

Reply: We corrected "hygroscopic" to "hygroscopicity".

223-225 – please clarify these sentences

Reply: We clarified the last three sentences of the second paragraph in Sect. 4.2.2 as: "This is physically reasonable that the aqueous droplet of these organics with limited solubility can be considered as being composed of an effectively insoluble core with a saturated solution. The organic with higher water solubility would dissolve more and have a higher molar concentration in the saturated solution. The higher molar concentration in water activity, which would lead the particles to become more hygroscopic."

229-231 - this is a run-on sentence - split the sentence and clarify

Reply: The second sentence of the first paragraph in Sect. 4.2.3 was revised as: "In this study, we plotted our measured  $\kappa$  of the 23 organic compounds with their O:C ratios in Fig. 6, and for a wider atmospheric implication we compared them against previous results from different atmospheric environments."

234 - "arises"

Reply: We corrected "arise" to "arises".

242 - "good" not "well"

Reply: We changed "well" to "good".

246 – "to our'

Reply: We replaced "as our" to "to our".

251 - "This, on the other hand, indicates"

Reply: Yes, we replaced "on the other side" to "on the other hand".

253 – "whose constitute may be diverged" – this is unclear, rephrase

Reply: We rephrased the eleventh sentence of the second paragraph in Sect. 4.2.3 to "The use of a simplified average property (i.e., O:C ratio) to describe the hygroscopicity of ambient organics, whose constitute may be complex, is quite risky as compounds with similar O:C ratio may vary considerably in hygroscopicity."

263 – "groups"

Reply: We corrected "group" to "groups".

264 - "processes"

Reply: We corrected "process" to "processes".

285 – "A detailed description of the HTDMA implementation, "

Reply: Yes, we replaced this sentence to "A detailed description of the HTDMA implementation".

**References**

[revised manuscript text omitted]

**Answers to Referee #2**

The authors appreciate the critical and constructive comments that the reviewer has given, which will help us to improve the communication with other researchers for the understanding of HTDMA work. All the requested comments are replied and corresponding suggestions are addressed in the revised version of the manuscript.

**Major comments:**

The authors present hygroscopicity data for 23 organic compounds measured with an HTDMA instrument. They compare these data with two thermodynamic models (AIOMFAC and E-AIM) and conclude that the models do not represent the hygroscopicity well.

Unfortunately, the study is conceptually flawed. The experimental design allows studying the hygroscopic growth starting from dry conditions to elevated relative humidities. This way, as the authors illustrate in Fig. S2 for an inorganic salt (ammonium sulfate), they can only probe the properties of the aqueous system at relative humidities, beyond those corresponding to the solubility limit of each binary system. However, what is relevant for the atmospheric application, where an aerosol particle consist of a multitude of organic compounds, is the water activity (or hygroscopicity) of the liquid state, because the organics will not crystallize under realistic atmospheric conditions (e.g. Marcolli et al., 2004).

Reply: The reviewer thinks the organics will not crystallize under realistic atmospheric conditions and considers the experimental design, which we dried our aerosols and made them crystallize, is not atmospherically relevant and conceptually flawed.

However, some of the organics may crystallize under realistic atmospheric conditions. For instance, Chan et al. (2008) reported the efflorescence RH (ERH) is 55-59% for succinic acid, >85% for adipic acid, 51-53% for pimelic acid. Chan et al. (2005) reported the ERH for a series of amino acids is 67-70%, 53-55% and 72-74% for alanine, glycine and glutamine, respectively. These results suggested that these organic compounds will crystallize under relative high RHs, which is likely occur in the real atmosphere. Moreover, when organic aerosols mixed with inorganic components, crystallization was still observed in previous studies (Lei et al., 2014; Luo et al., 2020), though this process is composition-dependent. Furthermore, Pajunoja et al. (2015) used an aerosol bounce instrument (ABI) to directly characterize the phase state of different SOA generated in a flow tube reactor and found that most of the generated particles are solid or semisolid at intermediate RH, for example (

---

## Referee Report (RR1)

Review revised manuscript # acp-2021-486 by Shuang Han et al.: «Hygroscopicity of organic compounds as a function of organic functionality, water solubility, molecular weight and oxidation level»

I am sorry but I am still convinced that this manuscript needs major revision or be rejected and not be published in ACP.

The main concern in my initial review was that with the approach used by the authors, they only measure hygroscopic growth under subsaturated conditions for all compounds, which crystallize in a binary aqueous mixture. That holds for example for all dicarboxylic acids besides malonic acid. I argued that an organic atmospheric aerosol particle typically contains a multitude, thousands of compounds, which each single compound only being present in a small fraction. In such multi-compound mixtures, the organics do not crystallize as have been proven by Marcolli et al. (2004). What is needed to constrain or to compare to thermodynamic models are the data in the liquid, supersaturated (often called sub-cooled) state. If you are interested in a comparison of thermodynamic models in the subsaturate state, bulk experiments yield much more accurate data than HTDMA experiments.

The authors reply to these concerns that crystallization does occurs in binary mixtures, which is true, but irrelevant, if you want to constrain or compare to thermodynamic models as written above. The authors further argue that bouncing experiments with SOA particles show that particle are solid or semisolid at intermediate RH. This is correct, but these particles are not crystalline solids, but in an amorphous state. So, you cannot argue that solid means crystalline. An amorphous solid may be slow in taking up water (as your sucrose example) but will not show deliquescence. If you wait long enough, it will reach thermodynamic equilibrium even at low humidity.

They write in their reply: *In our study, we dried the aerosols to relatively quite low RH, ensuring the crystallization of the organic compounds that likely crystallize under real conditions. Thus, the measured hygroscopic growth at elevated RHs could represent their hygroscopic behavior under ambient conditions, forming a realistic dataset.*

I strongly disagree with this statement as explained above.

Second, the authors fail to compare their data with data available in the literature using different techniques. I tried in my initial review to illustrate this with the data by Salecedo et al. (2006). The authors did a comparison with their data now and provided this figure:

[Figure]

However, they do not include it in the revised manuscript. Why?

This is exactly what is needed: comparison to previous data (which is available for almost all compounds discussed in the manuscript!) and a critical discussion of such a comparison. In the example for malonic acid shown above, it is not trivial to judge whether the E-AIM fits the complete dataset better of AIOMFAC or UManSysProp. Nor is it clear whether the different data agree within experimental error. Presently, such a discussion is missing in the manuscript.

To show another example: In the figure below I plot the data of malic acid of the manuscript together with the data from Peng et al. (2001) and two UManSysProp simulations (I used the density from UManSysProp from Girolami (1994) for the coversion of the Peng et al. data to size growth):

[Figure]

You may argue that all data agree at high humidity (> 85 %), but it is obvious that there is an increasing deviation between both data sets at lower humidity. (This is definitely larger than any error in the density estimation may cause.) The UManSysProp simulation assuming AIOMFAC activities comes closer to the Peng et al. data while assuming ideality in UManSysProp is closer to the data of the manuscript. It is not clear which data are the correct ones. Are the authors growth factors at low RH limited by kinetic uptake limitations (as in sucrose) also for malic acid?

What is however obvious to me is that a comparison of all data in the manuscript with literature data is essential as well as a discussion of discrepancies of these datasets.

Third, I feel the authors did not describe the UManSysProp on-line model well enough. As far as I understand, it is based on AIOMFAC for calculating the water activity (or did the authors assumed it to be ideal?). Hence it is clear that the curves shown in Fig. 1a of the manuscript for the dicarboxylic acid describe the liquid state hygroscopicity and cannot compared to the data if the experiments are

done with crystalline particles. Again, a comparison would be possible only if the supersaturated liquid particle were studied experimentally.

There are several option for selections with the UManSysProp web-model, for example the method to estimate supersaturated density. The authors do not write which method they used nor why they selected a particular one.

I feel, given the considerable discrepancies between the data presented and other datasets, it is not appropriate to judge which thermodynamic model represents the data better as done by the authors for the dicarboxylic acids.

Fourth, I appreciate the authors address the issue of kinetic limitations to water uptake in their text now, but they do not show how significant the effect could be. In the reply to my question, they provided a nice figure, which I copy here and which in my eyes show the problem unambiguously. It nicely shows that at low RH with high viscosity of the sucrose the authors do not reach water equilibrium while at high RH with water acting as a plasticizer they do. This figure may also explain the differences in the malic acid figure above?

[Figure]

However, the author ones more do not include this figure in the revised manuscript. Why? This figure needs to be discussed together with the added text!

Overall, I conclude that the manuscript requires at least a major revision with a comparison of all data to the data available in the literature, so that a reader sees the discrepancies and are not left with the impression that there are no other data available and also clearly see the limitations of the technique the authors have been using.

---

## Author Response (AR2)

**Answers to Referee #2**

The authors appreciate the critical comments that the reviewer has raised for the revised manuscript. The authors replied all the four major comments and made proper actions to the revised manuscript according to the suggestions by the reviewer.

**Major comments:**

The main concern in my initial review was that with the approach used by the authors, they only measure hygroscopic growth under subsaturated conditions for all compounds, which crystallize in a binary aqueous mixture. That holds for example for all dicarboxylic acids besides malonic acid. I argued that an organic atmospheric aerosol particle typically contains a multitude, thousands of compounds, which each single compound only being present in a small fraction. In such multicompound mixtures, the organics do not crystallize as have been proven by Marcolli et al. (2004). What is needed to constrain or to compare to thermodynamic models are the data in the liquid, supersaturated (often called sub-cooled) state. If you are interested in a comparison of thermodynamic models in the subsaturate state, bulk experiments yield much more accurate data than HTDMA experiments. The authors reply to these concerns that crystallization does occurs in binary mixtures, which is true, but irrelevant, if you want to constrain or compare to thermodynamic models as written above. The authors further argue that bouncing experiments with SOA particles show that particle are solid or semisolid at intermediate RH. This is correct, but these particles are not crystalline solids, but in an amorphous state. So, you cannot argue that solid means crystalline. An amorphous solid may be slow in taking up water (as your sucrose example) but will not show deliquescence. If you wait long enough, it will reach thermodynamic equilibrium even at low humidity.

Reply: We agree with the reviewer that comparison of current datasets with thermodynamic models might not be appropriate as the data in the liquid state is better to constrain these models. As the major purpose of current study is not to constrain thermodynamic models, we removed the part of comparison of our datasets with different thermodynamic models in our study, but focused more on the comparison of our data with previous data available measured by other studies. Future study measuring the hygroscopicity of compounds in the supersaturated state is seriously considered and carefully planned.

Considered the main concern raised by the reviewer, we made a new section discussing the major limitation of current instrument as well as some other bulk methods in Sect. 4. The aspect that different researchers should pay attention to when using HTDMA data was extended. Future work with combination of different methods was also suggested. The new section is listed below:

**4 Instrument limitations**

[revised manuscript text omitted]

Second, the authors fail to compare their data with data available in the literature using different techniques. I tried in my initial review to illustrate this with the data by Salecedo et al. (2006). The authors did a comparison with their data now and provided this figure: However, they do not include it in the revised manuscript. Why? This is exactly what is needed: comparison to previous data (which is available for almost all compounds discussed in the manuscript!) and a critical discussion of such a comparison. In the example for malonic acid shown above, it is not trivial to judge whether the E-AIM fits the complete dataset better of AIOMFAC or UManSysProp. Nor is it clear whether the different data agree within experimental error. Presently, such a discussion is missing in the manuscript together with the data from Peng et al. (2001) and two UManSysProp simulations (I used the density from UManSysProp from Girolami (1994) for the coversion of the Peng et al. data to size growth):

You may argue that all data agree at high humidity (> 85 %), but it is obvious that there is an

increasing deviation between both data sets at lower humidity. (This is definitely larger than any error in the density estimation may cause.) The UManSysProp simulation assuming AIOMFAC activities comes closer to the Peng et al. data while assuming ideality in UManSysProp is closer to the data of the manuscript. It is not clear which data are the correct ones. Are the authors growth factors at low RH limited by kinetic uptake limitations (as in sucrose) also for malic acid? What is however obvious to me is that a comparison of all data in the manuscript with literature data is essential as well as a discussion of discrepancies of these datasets.

Reply: The major suggestion here from the reviewer was to compare our data with data available in literature using different techniques. This was systematically done in the revised manuscript with all the corresponding figures replotted. As the comparison with the thermodynamic models was removed, no judgement about which model fits our results better was given. Detailed discussion for the comparison is listed below for each studied compound. Moreover, we carefully examined their available DRH and ERH values and made thorough discussions on their phase state. A new table (Table 2 in the revised manuscript) was made to summarize all these available information for a better review.

**3.1 Hygroscopicity of individual organics**

In this section, we summarized the measured hygroscopic properties of the 23 organic species, which are classified into three groups based on their functionality. Particles at the dry size of 200 nm were selected for analysis. Comparison with previous literature data for those compounds were systematically conducted in the following section. The available information with respect to the deliquescence RH (DRH), efflorescence RH (ERH), phase transition, the measured  $\kappa$  at 90% RH for each compound as well as the used instrument in different works are summarized in Table 2.

**3.1.1 Carboxylic acids**

Carboxylic acids are the most abundant water-soluble components identified in atmospheric aerosols (Chebbi and Carlier, 1996; Mochida et al., 2003; Kundu et al., 2010). Hygroscopic properties of straight-chain dicarboxylic acids have been extensively investigated in previous studies (Chan et al., 2008; Kuwata et al., 2013; Rickards et al., 2013), however, HTDMA data for dicarboxylic acids with additional substitutions and tricarboxylic acids are limited. To achieve a comprehensive overview of the hygroscopicity of carboxylic acids, we measured the water uptake of several common straight-chain dicarboxylic acids in the atmosphere and further extended the hygroscopic measurements for dicarboxylic acids with substitutions and tri-carboxylic acids. Figure 1 & 2 shows the measured humidograms of straight-chain dicarboxylic acids (Fig. 1a-f), dicarboxylic acids with substitutions (Fig. 2a-c) and tricarboxylic acids (Fig. 2d-f), respectively.

Among the studied straight-chain dicarboxylic acids, only malonic acid showed continuous hygroscopic growth with increasing RH and the measured GF at 90 % RH was 1.47, shown in agrees quite well with previous studies using other HTDMA systems or other Fig. 1a. This techniques, for instance, EDB, with longer residence time for humidification (Peng et al., 2001: Prenni et al., 2001: Wise et al., 2003: Salecedo et al., 2006: Jing et al., 2016). Previous studies (Swietlicki et al; 2008; Duplissy et al., 2009; Wu et al., 2011; Suda et al., 2013) suggested that the residence time for humidification may also potentially influence the observed water uptake of particles as the measured particles, especially for some organic compounds, may not reach their equilibrium humidified sizes during a quite short time of wetting. However, current results with good consistency with other studies of different residence time suggests that malonic acid may already reach equilibrium of humidification in our system. On the other hand, Braban et al. (2003) reported that the deliquescence RH (DRH) for malonic acid ranges between 69% - 91% RH. However, within this RH range we did not observe any clear evidences for phase transition and our results followed quite well with the ones obtained from the dehydration process by Peng et al. (2001), which further indicated that the measured malonic acid particles in our study existed at liquid state.

The other straight-chain dicarboxylic acids (i.e., **succinic, adipic, suberic and azelaic acids**) did not show any water uptake at RH <= 90 %, as shown in Fig. 1b-e. This non-hygroscopicity was also found for succinic acid in Peng et al. (2001), for adipic acid in Chan et al. (2008) and Dinar et al. (2008) and for azelaic acid in Chan et al. (2008) and Cook et al. (2011), either using a different HTDMA or an EDB. It has to be noted that no previous data was available for suberic acid, suggesting that other measurements using either similar or different methods should be performed for future comparison. Chan et al. (2008) explained that these dicarboxylic acids have quite low-solubility in water and once they crystallized, they

would not deliquesce even under high RH conditions (e.g., RH < 90 %). Moreover, we found that the measured GFs of azelaic and suberic acids were less than 1, which could be attributed to the adsorption of a small amount of water at the particle surface, leading to the rearrangements of the microstructure and compaction of the particle (Mikhailov et al., 2004; Mikhailov et al., 2009). It is interesting to note that we observed a slightly weak hygroscopicity for **pimelic acid** (Fig. 1f), while Chan et al. (2008) reported that pimelic acid particles was non-hygroscopic at RH < = 90%, and thus their DRH would be even higher. 
[revised manuscript text omitted]

For **xylitol and arabitol**, good agreement in the measured GF in general was found between our study and the ones by Bilde et al. (2014), though presented in an AGU abstract, with a small deviation in GF for arabitol at relatively high RH. According to the measured GF, both compounds were considered as more hygroscopic and likely existed as liquid state under dry conditions. From the measured humidogram, **mannitol** with GF less than unity at 90%RH is non-hygroscopic, which also suggested by previous literature (Ohrem et al., 2014; Martău et al., 2020). It has to be noted that no practical data were presented in their work and thus only our measured data was illustrated in Fig. 4f.

| <mark>Organic</mark>  | Compounds    | DRH (%)             | ERH (%)                    | Phase              | <mark>κ (90%</mark>   | Instruments             | References              |
|-----------------------|--------------|---------------------|----------------------------|--------------------|-----------------------|-------------------------|-------------------------|
| <mark>groups</mark>   |              |                     |                            | transition         | RH)                   |                         |                         |
|                       | Malonic      | <mark>n. o.</mark>  | •                          | <mark>n. o.</mark> | <mark>0.26</mark>     | HTDMA                   | This work               |
|                       | acid         | <mark>n. o.</mark>  | <mark>n. o.</mark>         | <mark>n. o.</mark> | <mark>0.29</mark>     | EDB                     | Peng et al., (2001)     |
|                       |              | <mark>n. o.</mark>  | <5                         | <mark>n. o.</mark> | <mark>0.52</mark>     | HTDMA                   | Prenni et al., (2001)   |
|                       |              | <mark>69-91</mark>  | 6                          | <mark>n. o.</mark> | ł                     | AFT-FTIR; SMC           | Braban et al., (2003)   |
|                       |              | <mark>71.9</mark>   | •                          | <mark>n. r.</mark> | 0.25                  | Nonisopiestic           | Wise et al., (2003)     |
|                       |              | <mark>70.9</mark>   | •                          | <mark>n. r.</mark> | <mark>0.28</mark>     | method                  | Salecedo et al., (2006) |
|                       |              | <mark>n. o.</mark>  | •                          | <mark>n. o.</mark> | <mark>0.39</mark>     | Nonisopiestic
method | Jing et al., (2016)     |
| Dicarboxylic
acids |              |                     |                            |                    |                       | HTDMA                   |                         |
|                       | Succinic     | <mark>n. o.</mark>  | <mark>n. o.</mark>         | <mark>n. o.</mark> | <mark>0.002</mark>    | HTDMA                   | This work               |
|                       | acid         | <mark>>90</mark> | <mark>55.2-59.</mark>
3 | <mark>n. o.</mark> | 0.002                 | EDB                     | Peng et al., (2001)     |
|                       | Adipic acid  | <mark>n. o.</mark>  | •                          | <mark>n. o.</mark> | <mark>-0.003</mark>   | HTDMA                   | This work               |
|                       |              | <mark>>90</mark> | <mark>>85</mark>        | <mark>n. o.</mark> | <mark>2.71 e⁻⁴</mark> | EDB                     | Chan et al., (2008)     |
|                       |              | <mark>98.5</mark>   | <mark>98.5</mark>          | <mark>n. o.</mark> | <mark>4.52 e⁻⁴</mark> | HTDMA                   | Dinar et al., (2008)    |
|                       | Azelaic acid | <mark>n. o.</mark>  | •                          | <mark>n. o.</mark> | -0.007                | HTDMA                   | This work               |

Table 2. Summary of the measured properties with respect to their DRH, ERH, phase transition, hygroscopicity parameter (90%RH) obtained in this work and previous literature.

|                            |              | <mark>>90</mark> | <mark>>85</mark> | <mark>n. o.</mark>       | 0.003              | EDB                     | Chan et al., (2008)      |
|----------------------------|--------------|---------------------|---------------------|--------------------------|--------------------|-------------------------|--------------------------|
|                            |              | <mark>81</mark>     | ł                   | <mark>n. o.</mark>       | ł                  | HTDMA                   | <mark>Cook (2011)</mark> |
|                            | Suberic acid | <mark>n. o.</mark>  | •                   | <mark>n. o.</mark>       | <mark>-0.03</mark> | HTDMA                   | This work                |
|                            |              | <mark>>90</mark> | <mark>>85</mark> | <mark>n. o.</mark>       | ł                  | EDB                     | Chan et al., (2008)      |
|                            | Pimelic acid | take up             | -                   | <mark>n. o.</mark>       | <mark>0.03</mark>  | HTDMA                   | This work                |
|                            |              | water>85            | 51.5-53             | <mark>n. o.</mark>       | 0.092 ±            | EDB                     | Chan et al., (2008)      |
|                            |              | <mark>>90</mark> |                     |                          | <mark>0.009</mark> |                         |                          |
|                            | Maleic acid  | <mark>n. o.</mark>  | -                   | <mark>n. o.</mark>       | 0.30               | HTDMA                   | This work                |
|                            |              | <mark>71-86</mark>  | <mark>48-51</mark>  | Observed          | <mark>0.28</mark>  | EDB                     | Choi and Chan (2002)     |
|                            |              | <mark>88.9</mark>   | •                   | <mark>n. o.</mark>       | <mark>0.27</mark>  | Nonisopiestic           | Wise et al., (2003)      |
| Dicarboxylic
acids with |              | <mark>89.1</mark>   |                     | <mark>n. r.</mark>       | •                  | <mark>method</mark>     | Marcolli et al., (2004)  |
|                            |              | <mark>n. o.</mark>  | •                   | <mark>n. o.</mark>       | <mark>0.23</mark>  | Nonisopiestic
method | Suda et al., (2013)      |
|                            |              |                     |                     |                          |                    | HTDMA                   |                          |
| substitutions              | DI-malic     | a leap at           |                     | Possible                 | 0.22               | HTDMA                   | This work                |
|                            | acid         | 80-85               | •                   | partial                  |                    |                         |                          |
|                            |              |                     |                     | deliquescence            |                    |                         |                          |
|                            |              | 75.00               | H                   | <mark>n. r.</mark>       | H                  | Nonisopiestic           | Apelblat et al., (1995)  |
|                            |              | /5-82               | <mark>n. o.</mark>  |
  <mark>n. o.</mark> | <mark>0.19</mark>  | method                  | Peng et al., (2001)      |
|                            |              | <mark>n. o.</mark>  |                     |                          |                    | EDB                     | Marcolli et al., (2004)  |
|                            |              |                     |                     | n. r.             |                    | Nonisopiestic    |                          |

|                        |                                        | 80.5
78.6                     | ł                    | <mark>n. r.</mark>               | ł                              | method
Nonisopiestic
method                | Clegg and Seinfeld
(2006)                                               |
|------------------------|----------------------------------------|----------------------------------|----------------------|----------------------------------|--------------------------------|--------------------------------------------------|----------------------------------------------------------------------------|
|                        | Tartaric acid                          | n. o.
77-78
n. o.
n. o. | -
-
n. o.
- | n. o.
n. r.
n. o.
n. r. | 0.22
-
0.19
-         | HTDMA
Nonisopiestic
method
EDB
HTDMA | This work
Apelblat et al., (1995)
Peng et al., (2001)
Cook (2011) |
| Tricarboxylic
acids | Citric acid
Cis-aconitic
acid    | n. o.
n. o.
>90
n. o.   | -
n. o.
-      | n. o.
n. o.
n. o.
n. o. | 0.18
0.18
0.20
0.21   | HTDMA
EDB
HTDMA
HTDMA                   | This work
Peng et al., (2001)
Wang et al., (2018)
This work       |
|                        | Butane-1,2,4-
tricarboxylic
acid | <mark>n. o.</mark>               | 1                    | <mark>n. o.</mark>               | 0.14                           | HTDMA                                            | This work                                                                  |
| Amino acids            | DL-alanine                             | n. o.
96.9                    | 67.3-70.
8        | n. o.
n. o.                   | -0.009
6.11 e -4 | HTDMA
EDB                                     | This work
Chan et al., (2005)                                           |
|                        | Glycine                                | <mark>n. o.</mark>               |                      | <mark>n. o.</mark>               | <mark>0.04</mark>              | HTDMA                                            | This work                                                                  |

|                          |                      | <mark>93.2</mark>   | <mark>53.6-55.</mark> | <mark>n. o.</mark> | <mark>0.04</mark>  | EDB      | Chan et al., (2005)              |
|--------------------------|----------------------|---------------------|-----------------------|--------------------|--------------------|----------|----------------------------------|
|                          |                      | <mark>60</mark>     | 2                     | not clear from     | •                  | ATR-FTIR | Darr et al., (2018)              |
|                          |                      |                     | <mark><35</mark>   | data               |                    |          |                                  |
|                          | L-Aspartic           | <mark>n. o.</mark>  | -                     | <mark>n. o.</mark> | <mark>0.14</mark>  | HTDMA    | This work                        |
|                          |                      | <mark>n. o.</mark>  | <mark>n. o.</mark>    | <mark>n. o.</mark> | <mark>0.18</mark>  | EDB      | Hartz et al., (2006)             |
|                          | L-Glutamine          | <mark>n. o.</mark>  | -                     | <mark>n. o.</mark> | <mark>0.16</mark>  | HTDMA    | This work                        |
|                          |                      | <mark>98.8</mark>   | <mark>72.1-74.</mark> | <mark>n. o.</mark> | <mark>0.006</mark> | EDB      | <mark>Chan et al., (2005)</mark> |
|                          |                      |                     | 2                     |                    |                    |          |                                  |
|                          | L-Serine             | <mark>n. o.</mark>  | -                     | <mark>n. o.</mark> | <mark>0.19</mark>  | HTDMA    | This work                        |
|                          |                      | <mark>99.1</mark>   | <mark>27.6-30.</mark> | <mark>n. o.</mark> | <mark>0.002</mark> | EDB      | <mark>Chan et al., (2005)</mark> |
|                          |                      |                     | 8                     |                    |                    |          |                                  |
|                          | Fructose      | <mark>n. o.</mark>  | -                     | <mark>n. o.</mark> | <mark>0.23</mark>  | HTDMA    | This work                        |
|                          |                      | <mark>n. o.</mark>  | <mark>n. o.</mark>    | <mark>n. o.</mark> | 0.18 ±             | EDB      | <mark>Chan et al., (2008)</mark> |
|                          |                      |                     |                       |                    | <mark>0.017</mark> |          |                                  |
| Sugar and sugar alcohols | <mark>Sucrose</mark> | <mark>n. o.</mark>  | -                     | <mark>n. o.</mark> | <mark>0.11</mark>  | HTDMA    | This work                        |
|                          |                      | <mark>>90</mark> | ł                     | <mark>n. o.</mark> | <mark>0.11</mark>  | DASH-SP  | Hodas et al., (2015)             |
|                          |                      | <mark>n. o.</mark>  | <mark>n. o.</mark>    | <mark>n. o.</mark> | <mark>0.11</mark>  | HTDMA    | Estillore et al., (2017)         |
|                          |                      | H                   | H                     | <mark>n. r.</mark> | <mark>0.10</mark>  | HTDMA    | Dawson et al., (2020)            |
|                          | Mannose              | <mark>n. o.</mark>  | -                     | <mark>n. o.</mark> | <mark>0.14</mark>  | HTDMA    | This work                        |

|            | <mark>n. o.</mark> | <mark>n. o.</mark> | <mark>n. o.</mark> | 0.183 ±
0.017  | EDB   | Chan et al., (2008)  |
|------------|--------------------|--------------------|--------------------|-------------------|-------|----------------------|
| Xylitol    | <mark>n. o.</mark> | •                  | <mark>n. o.</mark> | 0.21              | HTDMA | This work            |
|            | <mark>n. o.</mark> | ł                  | <mark>n. o.</mark> | <mark>0.18</mark> | HTDMA | Bilde et al., (2014) |
|            |                    |                    |                    |                   |       |                      |
| L-arabitol | <mark>n. o.</mark> | -                  | <mark>n. o.</mark> | <mark>0.19</mark> | HTDMA | This work            |
|            | <mark>n. o.</mark> | ł                  | <mark>n. o</mark>  | ł                 | HTDMA | Bilde et al., (2014) |
| D-mannitol | <mark>n. o.</mark> |                    | <mark>n. o.</mark> | -0.01             | HTDMA | This work            |
|            |                    |                    |                    |                   |       |                      |

 $\kappa$  values were derived from hydration data at around of 90% RH.

n. o. refers to no crystallization was observed.

n. r. refers to not reported in the work.

- refers to not measured in the work.

Figure 1. The hygroscopic growth values of straight-chain dicarboxylic acids particles (200 nm) measured in this study and their individual comparison with previous studies.